# Know Your Neighbors: Subgraph Importance Sampling for Heterophilic Graph Active Learning

## Abstract

Graph neural networks (GNNs) have shown superiority in various data mining tasks but rely heavily on extensively labeled nodes. To improve the training efficiency and select the most valuable nodes as the training set, graph active learning (GAL) has gained much attention. However, previous GAL methods are designed for homophilic graphs, and their effectiveness on heterophilic graphs is less examined. In this paper, we study active learning on heterophilic graphs, where nodes with the same labels are less likely to be connected. We are surprised to find that *previous GAL methods fail to outperform the naive random sampling on heterophilic graphs*. Through an insightful investigation, we find that previous GAL-selected training sets imply homophily even on heterophilic graphs, leading to their defectiveness. To address this issue, we propose the principle of *"Know Your Neighbors"* and design an active learning algorithm KyN specifically for heterophilic graphs. The primary idea of KyN is to let GNNs receive a correct homophily distribution by labeling nodes along with their neighbors. We build KyN based on subgraph sampling with probabilities proportional to $\ell_1$ Lewis weights, which has a solid theoretical guarantee. The effectiveness of KyN is evaluated on various real-world datasets.

## 1 Introduction

Graphs are ubiquitous in real-world applications, from recommendation system (Ma et al., 2024; Ni et al., 2024) and misconduct detection (Tao et al., 2024; Wu & Hooi, 2023) to AI for science (Gasteiger et al., 2021; Lam et al., 2023). Recently, graph neural networks (Kipf & Welling, 2017; Wu et al., 2019a; Velickovic et al., 2018; Chen et al., 2020) have become the de facto standard used in many graph learning tasks. Like other deep learning methods, the success of GNNs largely depends on the existence of high-quality training labels, and data labeling for these node samples is costly due to its reliance on human labor. To address this challenge, graph active learning has emerged as an effective approach for improving data efficiency (Song et al., 2023; Zhang et al., 2022a). GAL methods aim to maximize model performance by identifying the most informative nodes for annotation within a given labeling budget. Despite their success, we are surprised to find that previous GAL methods are only examined on homophilic graphs, i.e., nodes with the same labels are more likely to be connected. As heterophilic graph learning becomes a popular research direction, it is intriguing to ask:

> *Do graph active learning methods work with heterophily?*

The answer to this question is, unfortunately, no. We examined the performance of representative GAL methods on the heterophilic graph dataset, Roman-empire (Platonov et al., 2023b). The results are presented in Figure 1. The essential requirement of GAL methods is to consistently outperform the uniformly random sampling, since they are not very likely to be faster or simpler. However, we observe that none of the GAL methods can fulfill such requirements on this graph. Overall, uniformly random sampling might even be the strongest approach! These GAL methods work well on homophilic graphs, so this finding is highly unexpected.

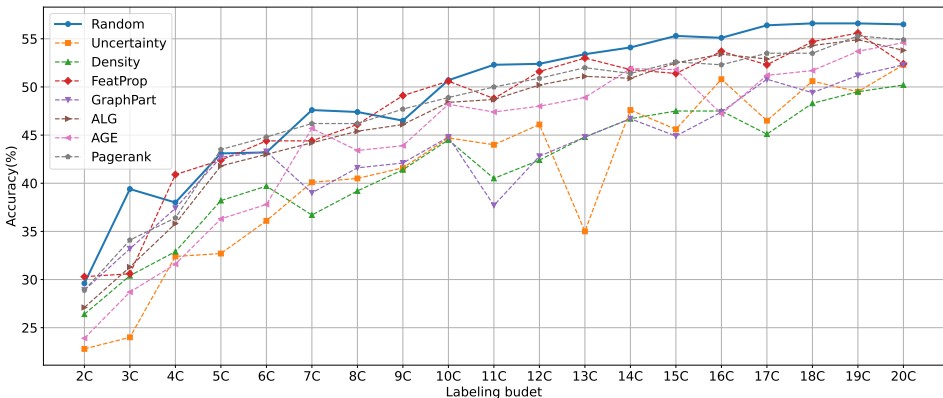

Figure 1: The performance of different GAL methods on the heterophilic graph datasets, Roman-empire. We set the labeling budget from $2C$ to $20C$, where $C$ is the number of classes in this dataset.

The defectiveness of GALs on heterophilic graphs is not only a new issue, but also a fatal one. According to recent surveys (Luan et al., 2024), heterophilic graphs are prevalent in many real-world applications, e.g., fraud/anomaly detection (Gao et al., 2023) and graph clustering (Pan & Kang, 2023). If one adopts off-the-shelf GAL methods on these graphs without knowing their pitfall, lots of time and human resources will be wasted. On the other hand, GNNs are known to be less optimal for heterophilic graphs, sometimes even outperformed by graph-agnostic models (Loveland et al., 2023). The last thing we want is a GAL method that is worse than random sampling to fuel the flames.

In this paper, we aim to develop an active learning algorithm for heterophilic graphs to fill the gap. Our method is motivated by issues of previous GALs. Specifically, we find that (1) existing GALs fail to let GNNs "know" whether a graph is homophilic or heterophilic, since they cannot produce the correct local homophily distribution. And (2) the fusion of ego-embeddings and neighbor-embeddings on heterophilic graphs makes nodes less distinguishable. To address the above problems, we propose the principle of "**K**now **Y**our **N**eighbors" and dub the model as KyN. By selecting nodes along with their neighbors, KyN yields the correct local homophily distribution. We build KyN with a novel $\ell_1$ Lewis weights subgraph sampling. KyN has a solid theoretical guarantee and is evaluated against various baselines on real-world datasets.

Our contributions are summarized as follows:

- We uncover the unexpected failure of active learning methods on heterophilic graphs. While they show their power on homophilic graphs, they are outperformed by the naive random sampling on heterophilic graphs. To the best of our knowledge, this is the first paper to reveal this phenomenon.

- We propose a novel method called KyN for active learning on heterophilic graphs. Our method is well-motivated by existing issues of previous GALs. KyN select training nodes along with their neighbors to yield a correct local homophily distribution that reflects the true homophilic/heterophilic nature of graphs.

- We conduct comprehensive experiments that demonstrate the superior performance of KyN on the heterophilic GAL task.

## 2 PRELIMINARIES

**Notations.** Let $G = (V, E, \boldsymbol{X}, \boldsymbol{Y})$ be a simple graph with node set $V$ and edge set $E$. $\boldsymbol{X} \in \mathbb{R}^{|V| \times f}$ is the node feature matrix, where $f$ is the number of dimensions of each feature. $\boldsymbol{Y} \in \mathbb{R}^{|V| \times C}$ is the one-hot label matrix with $C$ classes. We use $\mathbf{x}_i$ to represent the feature vector of the $i$-th node and $y_i$ as its label. We can also use the adjacency matrix $\boldsymbol{A} \in \{0,1\}^{|V| \times |V|}$, where the $(i,j)$-th entry is 1 if and only if the $i$-th node and the $j$-th node are connected. A $k$-hop neighborhood of node $i \in V$, $N_k(i)$ denotes the subgraph induced by the nodes that are reachable within $k$-steps of $i$.

**Homophily of graphs.** Homophily is a graph property describing the tendency of edges to connect similar nodes (Platonov et al., 2023a). Throughout our paper, a graph is *homophilic* if the nodes with the same labels are more likely to be connected. And a graph is *heterophilic* if the nodes with the same labels are less likely to be connected. Many statistics can measure the degree of homophily of a graph. We will mainly use the following two definitions of homophily/heterophily from previous works (Loveland et al., 2023).

**Definition 2.1** (Global Homophily). *The global homophily of a graph is defined as:*

$$h = \frac{|\{(u, v) : (u, v) \in E \land y_u = y_v\}|}{|E|},$$ (1)

*where $y_u$ is the label of node $u$.*

**Definition 2.2** (Local Homophily). *The local homophily of a node $t$ is defined as:*

$$h_t = \frac{|\{(u, t) : u \in N_1(t) \land y_u = y_t\}|}{|N_1(t)|}.$$ (2)

Intuitively, global homophily describes the overall property of a graph, while local homophily focuses on the specific neighborhood of each node. Previous works (Mao et al., 2023; Loveland et al., 2023) show that crucial properties (e.g., the prediction accuracy of GNNs) vary across local homophily levels, highlighting the importance of zooming in and analyze the diversity of node neighborhood.

**Graph active learning.** Active learning algorithms aim to select a training set that maximizes the performance of the models trained on it. Specifically, let $\mathcal{A}_{\text{GAL}}$ be a certain GAL algorithm that takes a graph $G$ and a labeling budget $B$ as inputs, the GAL-selected training set is $V_{\text{train}} = \mathcal{A}_{\text{GAL}}(G, B)$ with $|V_{\text{train}}| = B$. We acquire the labels of $V_{\text{train}}$ from an oracle, then train a GNN with them. The performance of the trained GNN can be used to measure the quality of $V_{\text{train}}$, which in turn reflects the effectiveness of $\mathcal{A}_{\text{GAL}}$. Since previous researches show that GCN is not suitable for heterophilic graphs (Platonov et al., 2023b), we will use the SAGE-mean (Hamilton et al., 2017) as the GNN encoder to eliminate additional impacts.

## 3 METHODOLOGY

### 3.1 DO GAL-SELECTED TRAINING SETS "TELL" THEIR HETEROPHILY?

Before designing a GAL algorithm that works with heterophily, we first investigate why previous GALs perform poorly on these graphs. Conceptually, GNNs cannot "know" whether a graph is homophilic or heterophilic if the training set does not "tell" them about it. Thus it is natural to first investigate the homophily-related information contained in GAL-selected training sets $V_{\text{train}}$. As local homophily emerges as an important statistic for heterophilic graphs, we can measure the homophily-related informativeness by the closeness between the distribution of local homophily of the training sets and that of the whole graph (i.e., the ground truth). Specifically, let $\mathcal{P}_{h_t}(G)$ be the local homophily distribution of the whole graph that can be empirically estimated with $\boldsymbol{A}$ and $\boldsymbol{Y}$. And let $\mathcal{P}_{h_t}(G_{\text{train}})$ be the local homophily distribution of a training set. Since we only know the labels of nodes in this training set, the distribution received by GNNs should be estimated with $\boldsymbol{A}_{\text{train}}$ and $\boldsymbol{Y}_{\text{train}}$, where $\boldsymbol{A}_{\text{train}}$ is the subgraph induced by the training set and $\boldsymbol{Y}_{\text{train}}$ is the labels of the training set. In other words, we do not count the unlabeled neighbors when investigating the homophily distribution of GAL-selected training sets. For some statistical distance $\mathcal{D}$, a GAL-selected training set that correctly contains homophily-related information should have small $\mathcal{D}(\mathcal{P}_{h_t}(G), \mathcal{P}_{h_t}(G_{\text{train}}))$, since it reflects the real distribution of local node homophily. We use the kernel density estimation to approximate the homophily distribution of each GAL-selected training set and present them in Figure 2.

Note that in our definition, the node $t$ itself is contained in its neighborhood $N_1(t)$, so each node will have at least one homophilic neighbor. Even under this setting, the ground truth of local homophily distribution is still right-skewed, indicating the heterophilic nature of the Roman-empire dataset. However, we observe that *previous GAL methods select training sets that show homophilic properties on this heterophilic dataset,* i.e., the distributions are left-skewed. It is then clear why GNN trained on these labeled sets fails to produce a satisfactory result: the model is given wrong,

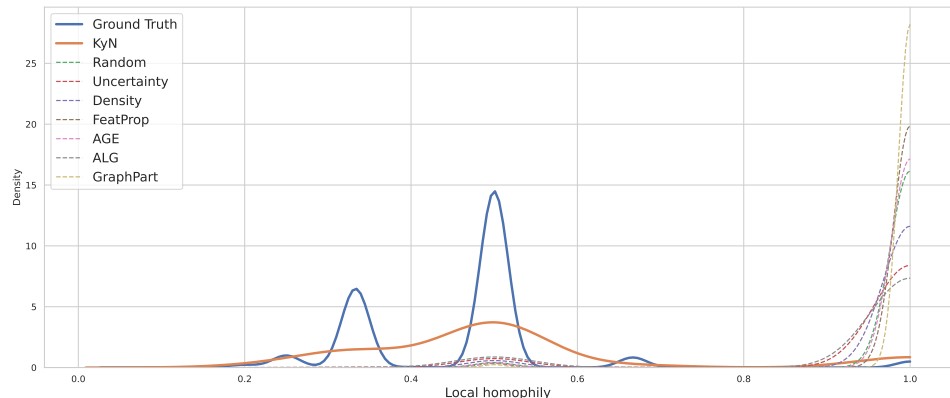

Figure 2: The local node homophily distribution plot of different GAL-selected training sets and that of the ground truth on the Roman-empire dataset. For a clear comparison, we also include our algorithm KyN. It is clear that KyN is the most similar to the ground truth distribution, and the only heterophilic one. We clip the distributions at 0 and 1. The labeling budget is $20C$.

even opposite, information in the first place. Formally, we show that a correct local homophily is actually necessary for a high accuracy. More details are in the Appendix A.

**Proposition 3.1.** *For predictions $\hat{\mathbf{y}} = \{\hat{y}_1, \cdots, \hat{y}_n\}$, let $\text{Acc} = \sum_{i=1}^{n} \mathbb{1}(y_i = \hat{y}_i)/n$, where $\mathbb{1}(\cdot)$ is the indicator function, let the accuracy of the ego-graph of node $i$ be*

$\text{Acc}_i = \sum_{j \in N(i)} \mathbb{1}(y_j = \hat{y}_j)/|N(i)|$, *and measure the correctness of local homophily with* $\mathcal{D}(\mathbf{h}, \hat{\mathbf{h}}) = \frac{1}{n} \sum_{i=1}^{n} \frac{1}{|N(i)|} |\sum_{j \in N(i)} \mathbb{1}(y_j = y_i) - \sum_{j \in N(i)} \mathbb{1}(\hat{y}_j = \hat{y}_i)|$, *where $\mathbf{h} = \{h_i\}_{i \in V}$ is the vector of local homophily, and $N(i)$ is 1-hop neighborhood of node $i$, we omit the subscript for the simplicity of notations. We have that a correct label homophily (i.e., small $\mathcal{D}(\mathbf{h}, \hat{\mathbf{h}})$) is necessary for high accuracy. Formally,*

$$\mathcal{D}(\mathbf{h}, \hat{\mathbf{h}}) \leq \frac{1}{n} \sum_{i=1}^{n} (1 - \text{Acc}_i) + (1 - \text{Acc}). \tag{3}$$

But why do previous GALs select homophilic training sets on a heterophilic graph? We argue that these GALs query nodes without their neighbors, leading to many isolated nodes in the induced subgraph. When fed to GNNs, these isolated nodes are viewed as strongly homophilic nodes (i.e., $h_t = 1$) since they are the only labeled nodes in their own neighborhood, resulting in an inaccurate homophily distribution. Therefore, the solution is rather straightforward: for heterophilic graphs, we should label their neighbors together with the selected nodes, which embodies the principle of "know your neighbors".

**Theorem 3.2** (The principle of "know your neighbors"). *For any labeled node $i$ in a graph $G$, the more its neighbors are known, the more accurate the estimate of local homophily will be. Formally, suppose we query $n_i$ node, then $\forall \epsilon \in (0, h_i)$,*

$$\mathcal{P}(|\hat{h}_i - h_i| \geq \epsilon) \leq 2 \exp(-2\epsilon^2 n_i), \tag{4}$$

*where $\hat{h}_i$ is the estimated local homophily of node $i$ and $h_i$ is the ground truth.*

### 3.2 SUBGRAPH IMPORTANCE SAMPLING

There are two methods to "know your neighbors":

- Sample then select $k$-hop: This approach first samples nodes with some GAL methods, then selects the $k$-hop neighbors of each node.
- Partition then sample: This approach first partitions the graph into disjoint subgraphs, and selects subgraphs with some GAL methods.

Figure 3 shows these two methods on a toy example. Due to neighbor explosion, the "sample then select $k$-hop" scheme tends to select a huge connected component, while "partition then sample" usually produces reasonably diverse subgraphs. Consider two nodes $u$ and $v$ are selected in the first stage of "sample then select one-hop", where $v$ is in the $k$-hop neighborhood of $u$, i.e., $v \in N_k(u)$. In the second stage, the union neighborhood of these two nodes will be gigantic, draining the labeling budget. Therefore, we design our algorithm within the "partition then sample" scheme. We introduce the details of each phase separately.

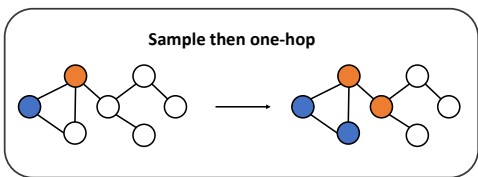

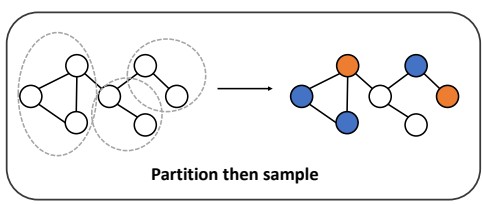

Figure 3: A toy example to illustrate two approaches to "know your neighbors". The colored nodes are labeled and will be used for training GNNs. We set $k = 1$ in the "sample then select $k$-hop" scheme.

**Partitioning.** We adopt the classic graph clustering algorithms METIS (Karypis & Kumar, 1998) to partition the graphs. For the graph $G$, we partition its nodes into $c$ groups: $V = \{V_1, V_2, \cdots V_c\}$, where $V_i$ is the $i$-th part. We then have $c$ subgraphs as

$$G_i = (V_i, E_i), \forall i \in [c], \quad (5)$$

where $E_i = \{(u, v) : (u, v) \in E \wedge u \in V_i \wedge v \in V_i\}$. Before moving on to the sampling phase, we need to generate a representation $\mathbf{R}_{G_i}$ for each subgraph $G_i$. It is possible to use naive readouts, e.g., take the average of node features:

$$\mathbf{R}_{G_i} = \frac{1}{|V_i|} \sum_{u \in V_i} \mathbf{x}_u. \quad (6)$$

However, since we are dealing with heterophilous graphs, Eq. (6) will lead to an inter-class fusion that makes subgraphs indistinguishable. Therefore, we propose a more sophisticated way to produce the representations. We first find a central node for each subgraph. The graph center is defined as follows:

**Definition 3.3** (Jordan center (Wasserman & Faust, 1994)). *The center of a graph is the set of all vertices of minimum eccentricity, i.e.,*

$$\arg\min_u \max_v d(u, v), \quad (7)$$

*where $d(\cdot, \cdot)$ is the geodesic distance.*

By finding a central node $n_c$, we are able to view the subgraph $G_i$ as an ego-graph centered at $n_c$. We can then separate the ego-embedding and neighbor-embeddings to yield a reasonable subgraph representation $\mathbf{R}_{G_i}$. This separation is known to be effective on heterophilic graphs (Zhu et al., 2020). Specifically, we compute the representation as follows:

$$\mathbf{R}_{G_i} = \text{CONCAT}(\mathbf{x}_{n_c}, \frac{1}{|N_1(n_c)| - 1} \sum_{i \in N_1(n_c) \setminus \{n_c\}} \mathbf{x}_i), \quad (8)$$

where $\text{CONCAT}(\cdot, \cdot)$ is the concatenation function. Note that this readout can also serve as a proxy of GraphSAGE (Hamilton et al., 2017) without learnable parameters, which is one of the few basic GNN encoders that work with heterophily (Platonov et al., 2023b).

**Sampling.** The goal of the sampling phase is to approximate the training loss of all subgraphs with only a small fraction of them. We sample these subgraphs with probabilities proportional to their $\ell_1$ Lewis weights. The formal definition of $\ell_1$ Lewis weights is:

**Definition 3.4** ($\ell_1$ Lewis weights (Cohen & Peng, 2015)). *For any matrix $\mathbf{M} \in \mathbb{R}^{n \times f}$ the $\ell_1$ Lewis weights are the unique values $\tau_1(\mathbf{M}), \cdots, \tau_n(\mathbf{M})$ such that,*

$$\tau_i(\mathbf{M})^2 = \mathbf{m}_i^T (\mathbf{M}^T \mathbf{W} \mathbf{M})^\dagger \mathbf{m}_i, \quad (9)$$

*where $\mathbf{W}$ is the diagonal matrix with $1/\tau_1(\mathbf{M}), \cdots, 1/\tau_n(\mathbf{M})$ as its diagonal, and the dagger symbol represents the pseudoinverse.*

The $\ell_1$ Lewis weights sampling has solid theoretical guarantees. In practice, we let $\boldsymbol{M} = \boldsymbol{R}$, where $\boldsymbol{R} = (\boldsymbol{R}_{G_1}, \cdots, \boldsymbol{R}_{G_c})^T$ is the subgraph representation matrix. We compute and normalize the $\ell_1$ Lewis weights $\tau_1(\boldsymbol{M}), \cdots, \tau_c(\boldsymbol{M})$ and select subgraphs with these probabilities. Once a subgraph is selected, we query all nodes within the subgraph, achieving "know your neighbors". After the number of labeled nodes reaches the budget, we feed the training set to GNNs for parameter optimization.

### 3.3 THEORETICAL ANALYSIS

The $\ell_1$ Lewis weights sampling has solid theoretical guarantees. For linear classification, the $\ell_1$ Lewis weights sampling yields a relative error coreset (Mai et al., 2021). A type of binary classification loss called nice hinge function is considered in Mai et al. (2021):

**Definition 3.5** (Nice Hinge Function (Mai et al., 2021)). *A function $f : \mathbb{R} \to \mathbb{R}^+$ is an $(L, a_1, a_2)$-nice hinge function if for fixed constants $L$, $a_1$ and $a_2$,*

*(1) $f$ is L-Lipschitz;    (2) $|f(z) - \mathrm{ReLU}(z)| \leq a_1, \forall z;$    (3) $f(z) \geq a_2, \forall z \geq 0$,*

*where $\mathrm{ReLU}(\cdot)$ is the rectified linear unit.*

We extend the theory to our multi-class classification on graphs. We show that $\ell_1$ Lewis sampling on $\boldsymbol{R}$ gives a relative error coreset for the cross-entropy (CE) loss. Specifically, minimizing the objective function on this selected coreset (i.e., the training set $V_{\mathrm{train}}$ in experiments) will yield a near minimizer over all subgraphs. Considering that the GCN encoder is not suitable for heterophilic graphs, while SAGE-Mean performs stably on heterophilic graphs (Platonov et al., 2023b), we follow previous work and use SAGE-Mean as the encoder. For simplicity, we use a one-layer SAGE-Mean encoder, the results are similar on any multi-layer linear GNNs. We reformulate the CE loss to show it is also a $(1, \ln 2, \ln 2)$-nice hinge function. The detailed proof is deferred to Appendix C.

**Theorem 3.6.** *For a one-layer GNN encoder, the CE loss is given by $L(\beta) = -\sum_{i=1}^c \ln(p(y_i)|\boldsymbol{R}_{G_i}, \beta)$, where $\beta$ is the learnable parameter. For a set of sampling values $p_i$ with $\sum_{i=1}^c p_i = m$ and $p_i \geq \frac{\mathcal{C} \max(\tau_i(\boldsymbol{R}), 1/c) \cdot \mu(\boldsymbol{R})^2}{\epsilon^2}$ for all $i$, where $\mathcal{C} = a \cdot \max(1, L, a_1, 1/a_2)^{10} \cdot \ln(\frac{\ln(c \max(1, L, a_1, 1/a_2) \cdot \mu(\boldsymbol{R})/\epsilon)m}{\delta})$ and $a$ is a fixed constant, $\mu(\boldsymbol{R}) = \sup_{\beta \neq 0} \frac{||(\boldsymbol{R}\beta)^+||_1}{||(\boldsymbol{R}\beta)^-||_1}$. If the sampling matrix $\boldsymbol{S} \in \mathbb{R}^{m \times c}$ has each row chosen independently as the $i^{th}$ standard basis vector scaled by $1/p_i$ with probability $p_i/m$, then with probability at least $1 - \delta$, we have the following relative error coreset:*

$$\left| \sum_{i=1}^m [\boldsymbol{S}f(z)]_i - L(\beta) \right| \leq \epsilon \cdot L(\beta), \tag{10}$$

*where $\boldsymbol{S}$ has $m = \tilde{O}(\frac{f\mu(\boldsymbol{R})^2}{\epsilon^2})$ rows.*

Thus, assuming $\beta^* = \arg\min_\beta L(\beta)$, and $\tilde{\beta}$ is the minimizer of the weighted loss $\sum_{i=1}^m [\boldsymbol{S}f(Z)]_i$, we have $L(\tilde{\beta}) \leq \frac{1+\epsilon}{1-\epsilon} \cdot L(\beta^*)$. This shows that minimizing the objective function on the sampled subset of size $m$ can produce an approximation close to the minimizer over all subgraph, achieving the goal of subgraph sampling. On the other hand, selecting all nodes within each subgraph achieves "know your neighbors", revealing the degree of homophily of a graph. To ensure a fair comparison, we sample subgraphs until the number of labeling nodes exceeds the budget, and keep the first $B$ nodes in experiments.

## 4 RELATED WORK

**Active learning** is a classic research direction that aims to mitigate annotation expenses (Ren et al., 2021; Matsushita et al., 2018). It is studied in many fields and under different settings, including computer vision (Bengar et al., 2021; Kim et al., 2021), nature language processing (Zhang et al., 2022b; Margatina et al., 2023) and general deep learning (Huang et al., 2024b; Yan & Huang, 2018; Tang & Huang, 2022). In the graph realm, AGE (Cai et al., 2017) is one of the earliest works that measure the informativeness of nodes by combining centrality, density, and uncertainty. AN-RMAB (Gao et al., 2018) improves AGE by learning weights using reinforcement learning. ALG (Zhang et al., 2021a) considers both the importance and correlation via the effective reception field

maximization. FeatProp (Wu et al., 2019b) first propagates features and then employs a clustering algorithm on the propagated node features. GraphPart (Ma et al., 2023) further enhances FeatProp by applying it to each graph partition. DOCTOR (Song et al., 2023) is a GAL method based on the expected model change maximization. GreedyET (Huang et al., 2024c) treat GAL as the aggregation involvement maximization. Some other papers focus on different settings that fit certain applications, e.g., noise/soft label (Zhang et al., 2022a; 2021b; 2024), fairness (Han et al., 2024) and transfer learning (Hu et al., 2020).

**Graph neural networks under heterophily** is an emerging topic in the graph realm. In heterophilic graphs, the nodes with the same labels are not more, sometimes even less, likely to be connected. The fusion phase of ordinary GNNs in these diverse neighborhoods makes nodes indistinguishable, leading to unsatisfactory performance. Various model architectures are proposed to address this challenge. $H_2$GCN (Zhu et al., 2020) is an early work on heterophily identifying designs crucial to the heterophily setting. CPGNN (Zhu et al., 2021) models different levels of homophily using a learnable class compatibility matrix in the aggregation step. GPR-GNN (Chien et al., 2021) is the generalized PageRank-inspired architecture designed to adapt to different label patterns. FAGCN (Bo et al., 2021) adaptively integrates different signals in the process of message passing with a self-gating mechanism. GloGNN (Li et al., 2022) generates node embedding by aggregating information from global nodes in the graph. GGCN (Yan et al., 2022) learns degree corrections and signed messages based on a unified theoretical perspective for heterophily and oversmoothing. M2M-GNN (Liang et al., 2024) unveil some potential pitfalls of signed message passing and design a new scheme to address the problem of undesirable representation update for multi-hop neighbors and vulnerability against oversmoothing issues. UniFilter (Huang et al., 2024a) develop an adaptive heterophily basis, this basis is then integrated with the homophily basis to construct a universal polynomial basis. In our paper, we train the GraphSAGE (Hamilton et al., 2017) to evaluate the quality of GAL-selected training sets, since it is one of the few basic GNN encoders that work with heterophily (Platonov et al., 2023b). We want to point out that GAL methods can be used with all previously mentioned GNNs that designed for heterophilic graphs. We omit such combinations without loss of generality.

**Coreset** is a research field that is very close to active learning. The main difference between the two problems is that we have access to labels before training set selection, but many coreset methods do not use labels so that they can be used for active learning. There are sampling works that focus on $\ell_2$-regression (Drineas et al., 2006; Li et al., 2013; Cohen et al., 2015) and $\ell_1$-regression (Clarkson, 2005; Sohler & Woodruff, 2011; Clarkson et al., 2016). Recent works show that coresets with relative error can be constructed on bounded complexity data for the logistic loss and hinge loss (Munteanu et al., 2019; Mai et al., 2021). Sampling-based coreset methods are also used for fields of active learning, e.g., multiple deep models active learning (Huang et al., 2024b). To the best of our knowledge, this paper is the first to explore Lewis weight sampling for graph active learning.

## 5 EXPERIMENTS

### 5.1 EXPERIMENTAL SETUP

We first compare KyN with other GAL methods on various real-world datasets: Roman-empire, Amazon-ratings, Tolokers, and Minesweeper (Platonov et al., 2023b), Wisconsin and Texas (Pei et al., 2020). We set the labeling budget to $5C$, $10C$, and $20C$, where $C$ is the number of classes in each dataset. This setting is common in previous GAL research (e.g., (Han et al., 2024)).

According to previous research, the message-passing approach of GCN is not suitable for heterophilic graphs, while SAGE-Mean shows relatively stable performance (Platonov et al., 2023b). This is because SAGE-Mean allows the "negative-aggregation" by concatenating the ego-embedding and neighbor-embedding. Therefore, to eliminate any additional impact from the encoder, we used SAGE-Mean as the GNN encoder instead of GCN on heterophilic graphs. We also used some GNNs designed specifically for heterophilic graphs as backbones to do a small number of experiments. Since the conclusions are consistent, we mainly use SAGE-Mean for simplicity and readability. We also provide the formula of SAGE-mean for readers who are not familiar with this encoder:

$$\boldsymbol{h}_v^l = \sigma(\boldsymbol{h}_v^{l-1}\boldsymbol{W}_1^l + (\text{mean}_{u \in N(v)}\boldsymbol{h}_u^{l-1})\boldsymbol{W}_2^l). \tag{11}$$

For these graphs, we train a three-layer encoder to evaluate the quality of the selected training set. The number of epochs is $300$. The learning rate is $0.01$ and the weight decay is $5 \times 10^{-4}$. The number of hidden units is $64$. All results are averaged over 10 runs, and standard deviations are reported. A key hyperparameter of our framework is the number of groups $c$, which is set to [1500, 2000, 2500, 25, 2500, 30] for Roman-empire, Amazon-ratings, Tolokers, Wisconsin, Minesweeper, and Texas, respectively.

All experiments are implemented using Python and PyTorch Geometric. Experiments are conducted on a server with an NVIDIA A100 GPU (80 GB memory) and an Intel Xeon Sapphire Rapids 9462 CPU. More implementation details can be found in Appendix D.

## 5.2 EXPERIMENTAL RESULTS

Table 1: The experimental results of KyN and other graph active learning methods. We report the mean classification accuracy and standard deviation trained on the training set selected by each GAL. The best results are **bolded**.

| Dataset | Roman-empire | | | Amazon-ratings | | |
|---|---|---|---|---|---|---|
| Budget | $5C$ | $10C$ | $20C$ | $5C$ | $10C$ | $20C$ |
| Random | $43.1 \pm 2.9$ | $50.7 \pm 1.0$ | $56.5 \pm 0.8$ | $30.2 \pm 2.6$ | $30.7 \pm 1.5$ | $31.3 \pm 0.6$ |
| Uncertainty | $32.7 \pm 4.4$ | $44.7 \pm 3.0$ | $52.3 \pm 2.6$ | $30.6 \pm 2.9$ | $30.8 \pm 2.7$ | $31.4 \pm 1.1$ |
| Density | $38.2 \pm 3.3$ | $44.5 \pm 2.4$ | $50.2 \pm 2.2$ | $30.5 \pm 2.1$ | $30.9 \pm 2.2$ | $31.1 \pm 0.8$ |
| AGE | $36.3 \pm 2.8$ | $48.2 \pm 2.4$ | $54.6 \pm 1.6$ | $29.3 \pm 1.8$ | $30.2 \pm 2.5$ | $30.8 \pm 1.6$ |
| ALG | $41.8 \pm 2.3$ | $48.4 \pm 1.8$ | $53.8 \pm 1.5$ | $30.8 \pm 1.5$ | $31.0 \pm 1.5$ | $31.6 \pm 1.0$ |
| FeatProp | $42.4 \pm 1.0$ | $50.6 \pm 2.1$ | $52.4 \pm 1.7$ | $30.2 \pm 1.3$ | $30.3 \pm 1.5$ | $30.9 \pm 0.6$ |
| GraphPart | $42.7 \pm 1.6$ | $44.8 \pm 2.5$ | $52.3 \pm 1.9$ | $30.4 \pm 2.2$ | $31.0 \pm 1.4$ | $32.1 \pm 0.7$ |
| KyN | $\mathbf{44.8} \pm 2.4$ | $\mathbf{51.4} \pm 1.3$ | $\mathbf{57.5} \pm 1.4$ | $\mathbf{31.2} \pm 1.7$ | $\mathbf{31.3} \pm 1.1$ | $\mathbf{32.3} \pm 0.4$ |
| Ave. Improve. | 5.2 | 4.0 | 4.3 | 0.9 | 0.6 | 1.0 |

| Dataset | Tolokers | | | Wisconsin | | |
|---|---|---|---|---|---|---|
| Budget | $5C$ | $10C$ | $20C$ | $5C$ | $10C$ | $20C$ |
| Random | $65.4 \pm 3.9$ | $68.8 \pm 4.7$ | $69.0 \pm 3.2$ | $71.7 \pm 4.0$ | $78.6 \pm 3.3$ | $86.1 \pm 2.3$ |
| Uncertainty | $68.9 \pm 8.6$ | $71.4 \pm 8.0$ | $71.7 \pm 4.6$ | $71.6 \pm 5.9$ | $78.7 \pm 3.9$ | $88.1 \pm 2.1$ |
| Density | $62.7 \pm 9.2$ | $68.5 \pm 6.4$ | $68.6 \pm 4.2$ | $68.7 \pm 1.3$ | $72.5 \pm 1.1$ | $83.7 \pm 2.0$ |
| AGE | $66.6 \pm 7.8$ | $69.4 \pm 5.6$ | $70.9 \pm 4.7$ | $69.2 \pm 2.2$ | $78.2 \pm 0.7$ | $87.4 \pm 3.0$ |
| ALG | $67.3 \pm 6.4$ | $69.6 \pm 6.1$ | $70.8 \pm 4.3$ | $70.8 \pm 3.7$ | $78.5 \pm 3.2$ | $86.9 \pm 2.6$ |
| FeatProp | $62.3 \pm 7.1$ | $70.6 \pm 5.3$ | $66.8 \pm 3.9$ | $71.9 \pm 2.8$ | $78.8 \pm 1.7$ | $87.9 \pm 2.5$ |
| GraphPart | $69.8 \pm 6.8$ | $71.2 \pm 4.3$ | $71.5 \pm 4.1$ | $69.7 \pm 3.1$ | $78.9 \pm 1.5$ | $87.2 \pm 2.9$ |
| KyN | $\mathbf{71.0} \pm 4.5$ | $\mathbf{71.8} \pm 3.5$ | $\mathbf{72.9} \pm 4.2$ | $\mathbf{72.5} \pm 3.5$ | $\mathbf{79.1} \pm 1.1$ | $\mathbf{88.5} \pm 2.2$ |
| Ave. Improve. | 4.9 | 1.9 | 3.0 | 2.0 | 1.4 | 1.8 |

| Dataset | Minesweeper | | | Texas | | |
|---|---|---|---|---|---|---|
| Budget | $5C$ | $10C$ | $20C$ | $5C$ | $10C$ | $20C$ |
| Random | $72.9 \pm 5.2$ | $75.0 \pm 3.6$ | $77.1 \pm 3.1$ | $73.3 \pm 2.9$ | $82.6 \pm 3.0$ | $92.8 \pm 2.2$ |
| Uncertainty | $68.7 \pm 7.9$ | $75.4 \pm 6.2$ | $76.7 \pm 4.1$ | $73.4 \pm 2.9$ | $84.1 \pm 2.5$ | $\mathbf{94.7} \pm 1.4$ |
| Denstiy | $67.3 \pm 9.8$ | $73.0 \pm 7.9$ | $75.1 \pm 3.2$ | $73.5 \pm 2.5$ | $78.6 \pm 2.7$ | $91.8 \pm 1.6$ |
| AGE | $71.0 \pm 3.8$ | $75.7 \pm 3.8$ | $76.4 \pm 2.5$ | $74.3 \pm 2.3$ | $80.4 \pm 2.1$ | $89.3 \pm 0.7$ |
| ALG | $71.6 \pm 4.7$ | $75.5 \pm 5.4$ | $76.8 \pm 2.7$ | $74.6 \pm 2.7$ | $83.5 \pm 2.6$ | $91.3 \pm 1.1$ |
| FeatProp | $73.1 \pm 4.4$ | $75.6 \pm 2.9$ | $76.2 \pm 2.3$ | $76.2 \pm 2.8$ | $82.2 \pm 2.4$ | $92.9 \pm 1.5$ |
| GraphPart | $72.8 \pm 5.6$ | $75.9 \pm 3.1$ | $76.8 \pm 2.1$ | $77.1 \pm 2.4$ | $83.9 \pm 2.2$ | $92.7 \pm 1.9$ |
| KyN | $\mathbf{73.3} \pm 5.3$ | $\mathbf{76.5} \pm 3.6$ | $\mathbf{77.8} \pm 2.8$ | $\mathbf{77.4} \pm 2.9$ | $\mathbf{84.3} \pm 4.1$ | $93.2 \pm 1.6$ |
| Ave. Improve. | 2.2 | 1.3 | 1.4 | 2.8 | 2.1 | 1.0 |

**Performance on heterophilic graphs.** Table 1 shows the performance of GALs on heterophilic graphs. The results show that KyN achieves the best performance on all heterophilic graphs with different labeling budgets. As mentioned earlier, we observe that on many heterophilic datasets (e.g.,

Roman-empire and Minesweeper), previous GALs fail to consistently outperform the naive random sampling. The gap between previous GAL methods and random sampling can even reach as high as 10.4% and 5.6%. Compared to previous GAL methods, the performance improvement of KyN on six datasets can reach up to 12.1%, 1.9%, 8.7%, 6.6%, 6.0% and 5.7%, respectively. The success of KyN is due to the unveiling of the heterophilic nature by the selection training sets. As mentioned in Section 3.1 and Figure 2, previous GAL-selected training sets imply homophilic property even on heterophilic graphs. This is because these GALs are only designed for informativeness and coverage of graphs, not homophily. In contrast, we address this issue by the principle of "know your neighbors".

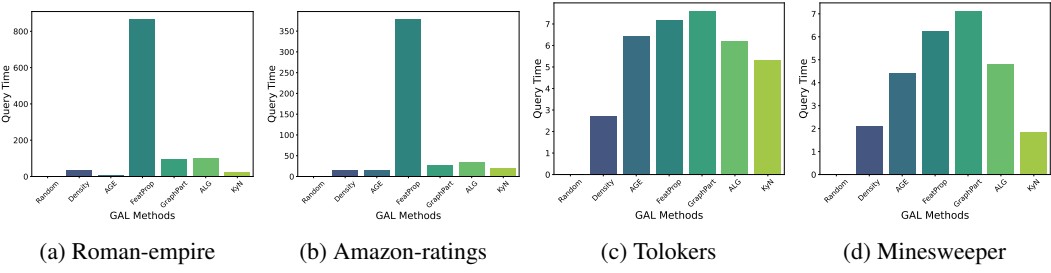

(a) Roman-empire     (b) Amazon-ratings     (c) Tolokers     (d) Minesweeper

Figure 4: The runtime (in second) comparison between KyN and other GALs.

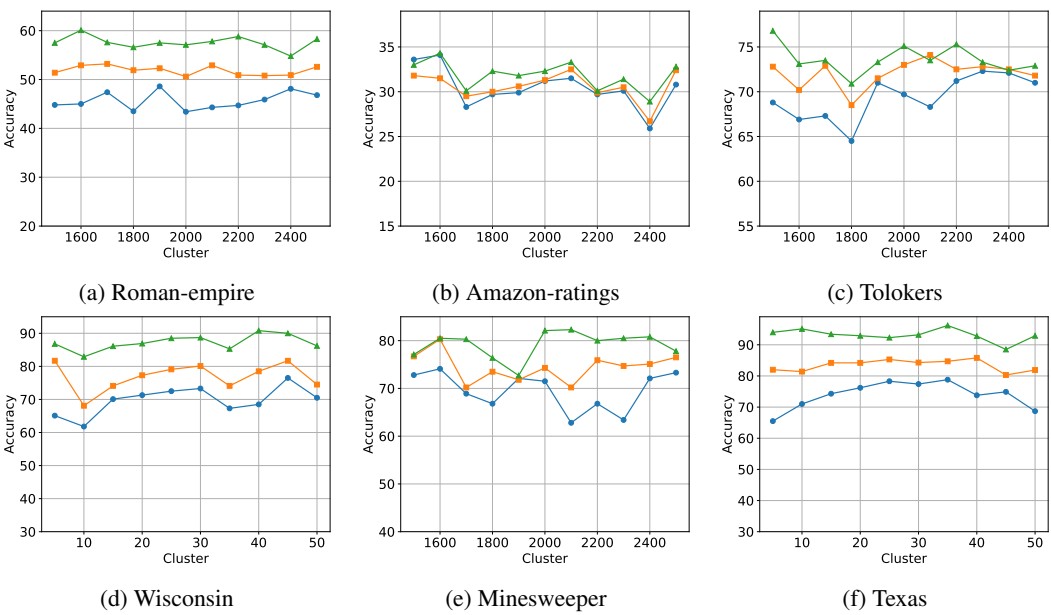

(a) Roman-empire     (b) Amazon-ratings     (c) Tolokers

(d) Wisconsin     (e) Minesweeper     (f) Texas

Figure 5: The hyperparameter sensitivity analysis of KyN. The green, orange, and blue lines are accuracy curves with labeling budgets of $5C$, $10C$, and $20C$, respectively.

**Runtime comparison.** Although the purpose of this paper is not to design an efficient GAL method, we still hope that KyN can achieve a reasonable runtime. For graph active learning, if the time resources consumed are excessive, then such a GAL method is not feasible for practical implementation. Figure 4 shows the runtime of each GAL method on relatively larger datasets. We observe that the runtime of KyN is acceptable considering its performance. On these datasets, it is faster than prevalent GAL methods, e.g., FeatProp, GraphPart, and ALG. On the Roman-empire and Amazon-ratings datasets, the time consumption of KyN is even an order of magnitude lower than that of FeatProp. Moreover, the runtime of KyN is negligible compared with human annotation.

**Hyperparameter sensitivity.** We study the influence of the number of groups $c$ on the performance of KyN. As mentioned earlier, we do not want a giant connected component as the training set, since it lacks diversity. So the bottom line is to keep the number of nodes in a subgraph under the labeling budget, i.e., $|V| \leq Bc$, where $B$ is the budget. On the other hand, a $c$ that is too large should also be avoided as it will not achieve our principle of "know your neighbors". Figure 5 shows the results on six datasets. We observe that KyN is robust to the choice of $c$. In practice, we recommend choosing $c$ around $\frac{|V|}{C}$, where $C$ is the number of classes. We also want to point out that some choice of $c$ will lead to a better performance than Table 1. This is normal since we do not tune $c$ with the final accuracy to avoid data leakage.

**Heterophilic GNNs as backbones.** We use two heterophilic GNNs, FAGCN (Bo et al., 2021) and M2M-GNN (Liang et al., 2024), as backbones to compare different GALs on the Roman-empire dataset. The results are presented in Table 2. We observe that KyN still achieve the best performance with these two backbones. In other experiments in this article, we stick to SAGE-Mean as the backbone so that readers who are not familiar with heterophilic GNN can understand it more easily.

**Performance on a large heterophilic graph.** To verify the scalability of KyN, we compare different GAL methods on a large heterophilic graph, snap-patents. This dataset contains more than two million nodes and thirteen million edges. The results are presented in Table 3. We observe that KyN achieves the best performance and the runtime is also reasonable. This experiment shows the efficiency of KyN.

Table 2: The experimental results with heterophilic GNNs as backbones on the Roman-empire dataset. The labeling budget is $20C$.

| Method | FAGCN | M2M-GNN |
|---|---|---|
| Random | $52.0 \pm 0.5$ | $58.3 \pm 1.3$ |
| Uncertainty | $47.7 \pm 1.8$ | $54.9 \pm 1.0$ |
| Density | $45.3 \pm 1.1$ | $51.5 \pm 1.2$ |
| AGE | $50.5 \pm 1.9$ | $55.7 \pm 1.4$ |
| ALG | $51.2 \pm 1.3$ | $56.3 \pm 1.2$ |
| FeatProp | $51.7 \pm 0.9$ | $57.0 \pm 0.8$ |
| GraphPart | $51.6 \pm 1.2$ | $56.1 \pm 0.8$ |
| KyN | $\mathbf{53.5} \pm 1.4$ | $\mathbf{59.2} \pm 1.0$ |

Table 3: The experimental results on a large heterophilic graph, snap-patents. The labeling budget is $5C$. We report the classification accuracy and runtime (in seconds). OOT (out-of-time) indicates the scenario where the algorithm failed to finish within 24 hours.

| Method | Accuracy | Runtime |
|---|---|---|
| Random | 32.9 | 0.06 |
| Uncertainty | 25.5 | 0.45 |
| Density | 25.1 | 752 |
| AGE | 23.7 | 3504 |
| ALG | OOT | - |
| FeatProp | 21.6 | 7245 |
| GraphPart | OOT | - |
| KyN | **33.7** | 651 |

**More detailed component analysis.** Due to the page limit, we defer ablation studies and other component analyses to Appendix H.

## 6 CONCLUSION

In this paper, we investigate a new research problem, heterophilic graph active learning. We observe that previous GAL methods that work perfectly on homophilic graphs fail to outperform naive random sampling on heterophilic graphs. Through an insightful investigation of the local homophily distribution, we find that previous GAL-selected training sets imply homophilic properties on heterophilic graphs. We argue that the previous design principle of informativeness and coverage on graphs will inevitably produce isolated training nodes that is harmful for heterophilic GALs. To address this issue, we propose a novel principle of "know your neighbors" and dub our model as KyN. KyN unveils the homophilic/heterophilic nature of graphs by labeling nodes along with their neighbors. We implement KyN with $\ell_1$ Lewis weights sampling, which has solid theoretical guarantees. Extensive experiments show the effectiveness of our method.

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

## A   PROOF OF PROPOSITION3.1

**Proposition A.1.** *For predictions $\hat{\mathbf{y}} = \{\hat{y}_1, \cdots, \hat{y}_n\}$, let*

$$\text{Acc} = \frac{1}{n}\sum_{i=1}^{n}\mathbb{1}(y_i = \hat{y}_i), \tag{12}$$

*where $\mathbb{1}(\cdot)$ is the indicator function, let the accuracy of the ego-graph of node $i$ be*

$$\text{Acc}_i = \frac{1}{|N(i)|}\sum_{j\in N(i)}\mathbb{1}(y_j = \hat{y}_j), \tag{13}$$

*and measure the correctness of local homophily with*

$$\mathcal{D}(\boldsymbol{h},\hat{\boldsymbol{h}}) = \frac{1}{n}\sum_{i=1}^{n}\frac{1}{|N(i)|}|\sum_{j\in N(i)}\mathbb{1}(y_j = y_i) - \sum_{j\in N(i)}\mathbb{1}(\hat{y}_j = \hat{y}_i)|, \tag{14}$$

*where $N(i)$ is 1-hop neighborhood of node $i$, we omit the subscript for the simplicity of notations. We have that a correct label homophily (i.e., small $\mathcal{D}(\boldsymbol{h},\hat{\boldsymbol{h}})$) is necessary for high accuracy. Formally,*

$$\mathcal{D}(\boldsymbol{h},\hat{\boldsymbol{h}}) \le \frac{1}{n}\sum_{i=1}^{n}(1 - \text{Acc}_i) + (1 - \text{Acc}). \tag{15}$$

*Proof.*

$$
\begin{aligned}
\mathcal{D}(\boldsymbol{h},\hat{\boldsymbol{h}}) &= \frac{1}{n}\sum_{i=1}^{n}\frac{1}{|N(i)|}|\sum_{j\in N(i)}\mathbb{1}(y_j = y_i) - \sum_{j\in N(i)}\mathbb{1}(\hat{y}_j = \hat{y}_i)| \\
&= \frac{1}{n}\sum_{i=1}^{n}\frac{1}{|N(i)|}\mathbb{1}(y_i = \hat{y}_i)|\sum_{j\in N(i)}\mathbb{1}(y_j = y_i) - \sum_{j\in N(i)}\mathbb{1}(\hat{y}_j = \hat{y}_i)| \\
&\quad + \frac{1}{n}\sum_{i=1}^{n}\frac{1}{|N(i)|}\mathbb{1}(y_i \ne \hat{y}_i)|\sum_{j\in N(i)}\mathbb{1}(y_j = y_i) - \sum_{j\in N(i)}\mathbb{1}(\hat{y}_j = \hat{y}_i)| \\
&\le \frac{1}{n}\sum_{i=1}^{n}\frac{1}{|N(i)|}\mathbb{1}(y_i = \hat{y}_i)\sum_{j\in N(i)}\mathbb{1}(y_j \ne \hat{y}_j) + \frac{1}{n}\sum_{i=1}^{n}\mathbb{1}(y_i \ne \hat{y}_i) \\
&\le \frac{1}{n}\sum_{i=1}^{n}(1 - \text{Acc}_i) + (1 - \text{Acc}).
\end{aligned}
\tag{16}
$$

$\square$

Theorem 3.1 shows that higher accuracy implies correct local homophily (i.e., small $\mathcal{D}(\boldsymbol{h},\hat{\boldsymbol{h}})$), and wrong local homophily (i.e., large $\mathcal{D}(\boldsymbol{h},\hat{\boldsymbol{h}})$) implies lower accuracy. This result further shows the importance of homophily to the behavior of GNNs. We also want to highlight that the left-hand side of Eq. (15) is a global measure, and the right-hand side is also a global measure since the first term is a summation over all nodes $i \in V$. So even if a node $j$ is not in the neighborhood of some node $i$, its accuracy still counts. The second term of the right-hand side is also a global measure, so there is no theoretical gap between the local and the global.

## B   PROOF OF THEOREM 3.2

**Theorem B.1** (The principle of "know your neighbors"). *For any labeled node $i$ in a graph $G$, the more its neighbors are known, the more accurate the estimate of local homophily will be. Formally, suppose we query $n_i$ node, then $\forall \epsilon \in (0, h_i)$,*

$$\mathcal{P}(|\hat{h}_i - h_i| \geq \epsilon) \leq 2\exp(-2\epsilon^2 n_i), \tag{17}$$

where $\hat{h}_i$ is the estimated local homophily of node $i$ and $h_i$ is the ground truth.

*Proof.* Suppose we have $K$ out of $n_i$ nodes that have the same label with $i$. We will estimate the local homophily use $\hat{h}_i = \frac{K}{n_i}$, and the ground truth is $h_i = \frac{P}{|N(i)|}$, where $P$ is the total number of positive neighbors. We observe that $K$ follows a hypergeometric distribution, $K \sim \mathrm{HG}(|N(i)|, P, n_i)$. Therefore, $\forall \epsilon \in (0, h_i)$,

$$
\begin{aligned}
\mathcal{P}(|\hat{h}_i - h_i| \geq \epsilon) &= \mathcal{P}(\hat{h}_i - h_i \geq \epsilon) + \mathcal{P}(\hat{h}_i - h_i \leq -\epsilon) \\
&= \mathcal{P}(K \geq (h_i + \epsilon)n_i) + \mathcal{P}(K \leq (h_i - \epsilon)n_i) \\
&\leq 2\exp(-2\epsilon^2 n_i).
\end{aligned}
\tag{18}
$$

$\square$

Theorem 3.2 shows that, for any node $i$, the more its neighbors are known, the more accurate the estimate of local homophily will be. Since we have $\mathcal{D}(\boldsymbol{h}, \hat{\boldsymbol{h}}) = \frac{1}{n}\sum_{i=1}^n |h_i - \hat{h}_i|$, small $|h_i - \hat{h}_i|$ is necessary for $\mathcal{D}(\boldsymbol{h}, \hat{\boldsymbol{h}})$. However, since we are working on a GAL setup with a limited budget, it is not possible to make all nodes $i \in V$ "know their neighbors". What we can do is to ensure that as many nodes as possible meet this principle. Besides, when node $i$ knows its neighbor $j$, it implies that $j$ also knows its neighbor $i$, so the process is reciprocal.

## C PROOF OF THEOREM 3.6

We will make use of the following result on linear classification with nice hinge function:

**Theorem C.1** (Nice Hinge Function – Relative Error Coreset (Mai et al., 2021)). *For some matrix $\boldsymbol{X} \in \mathbb{R}^{n \times d}$ and an $(L, a_1, a_2)$-nice hinge function $f$ and $a_2 > 0$. For a set of sampling value $p_i$ with $\sum_{i=1}^n p_i = m$ and $p_i \geq \frac{\mathcal{C}\max(\tau_i(\boldsymbol{X}), 1/n)\cdot\mu(\boldsymbol{X})^2}{\epsilon^2}$ for all $i$, where $\mathcal{C} = a \cdot \max(1, L, a_1, 1/a_2)^{10} \cdot \ln(\frac{\ln(n\max(1, L, a_1, 1/a_2)\cdot\mu(\boldsymbol{X})/\epsilon)m}{\delta})$ and $a$ is a fixed constant, if we generate $\boldsymbol{S} \in \mathbb{R}^{m \times n}$ with each row chosen independently as the $i^{th}$ standard basis vector times $1/p_i$ with probability $p_i/m$, then with probability at least $1 - \delta$, $\forall \beta \in \mathbb{R}^d$,*

$$|\sum_{i=1}^m [\boldsymbol{S}f(\boldsymbol{X}\beta)]_i - \sum_{i=1}^n f(\boldsymbol{X}\beta)_i| \leq \epsilon \cdot \sum_{i=1}^n f(\boldsymbol{X}\beta)_i, \tag{19}$$

*where $\boldsymbol{S}$ has $m = \tilde{O}(\frac{d\mu(\boldsymbol{X})^2}{\epsilon^2})$ rows.*

We extend the above theorem to our multi-class classification on GNNs.

**Theorem C.2.** *For a $1$-layer GraphSAGE encoder, the CE loss is given by $L(\beta) = -\sum_{i=1}^c \ln(p(y_i)|\boldsymbol{R}_{G_i}, \beta)$, where $\beta$ is the learnable parameter. For a set of sampling values $p_i$ with $\sum_{i=1}^c p_i = m$ and $p_i \geq \frac{\mathcal{C}\max(\tau_i(\boldsymbol{R}), 1/c)\cdot\mu(\boldsymbol{R})^2}{\epsilon^2}$ for all $i$, where $\mathcal{C} = a \cdot \max(1, L, a_1, 1/a_2)^{10} \cdot \ln(\frac{\ln(c\max(1, L, a_1, 1/a_2)\cdot\mu(\boldsymbol{R})/\epsilon)m}{\delta})$ and $a$ is a fixed constant, $\mu(\boldsymbol{R}) = \sup_{\beta \neq 0} \frac{||(\boldsymbol{R}\beta)^+||_1}{||(\boldsymbol{R}\beta)^-||_1}$. If the sampling matrix $\boldsymbol{S} \in \mathbb{R}^{m \times c}$ has each row chosen independently as the $i^{th}$ standard basis vector scaled by $1/p_i$ with probability $p_i/m$, then with probability at least $1 - \delta$, we have the following relative error coreset:*

$$\left|\sum_{i=1}^m [\boldsymbol{S}f(z)]_i - L(\beta)\right| \leq \epsilon \cdot L(\beta), \tag{20}$$

*where $\boldsymbol{S}$ has $m = \tilde{O}(\frac{f\mu(\boldsymbol{R})^2}{\epsilon^2})$ rows.*

*Proof.* Consider a single-layer GraphSAGE, where $\boldsymbol{R} = (\boldsymbol{R}_{G_1}, \dots, \boldsymbol{R}_{G_c})^T \in \mathbb{R}^{c \times d}$ is the representation matrix, where $d = 2f$ in our setting, $y \in \{1, \dots, C\}^c$ is the label vector, and

$\beta \in \mathbb{R}^{d \times k}$ is the parameter. The CE Loss is given by $L(\beta) = -\sum_{i=1}^{c} \ln(p(y_i|\boldsymbol{R}_{G_i}))$, where $p(y_i|\boldsymbol{R}_{G_i}) = e^{\boldsymbol{R}_{G_i}^T \beta_{y_i}} / \sum_{j=1}^{C} e^{\boldsymbol{R}_{G_i}^T \beta_j}$. We can reformulate $L(\beta)$ as:

$$
\begin{aligned}
L(\beta) &= -\sum_{i=1}^{c} \ln(p(y_i|\boldsymbol{R}_{G_i})) \\
&= -\sum_{i=1}^{c} \ln\left(\frac{e^{\boldsymbol{R}_{G_i}^T \beta_{y_i}}}{\sum_{j=1}^{C} e^{\boldsymbol{R}_{G_i}^T \beta_j}}\right) \\
&= \sum_{i=1}^{c} \ln\left(\frac{\sum_{j=1}^{C} e^{\boldsymbol{R}_{G_i}^T \beta_j}}{e^{\boldsymbol{R}_{G_i}^T \beta_{y_i}}}\right) \\
&= \sum_{i=1}^{c} \ln\left(1 + \frac{\sum_{j \neq y_i} e^{\boldsymbol{R}_{G_i}^T \beta_j}}{e^{\boldsymbol{R}_{G_i}^T \beta_{y_i}}}\right) \\
&= \sum_{i=1}^{c} \ln(1 + e^{z_i}) \\
&= \sum_{i=1}^{c} f(z)_i,
\end{aligned}
\tag{21}
$$

where we let $z \in \mathbb{R}^c$, and $z_i = \ln\left(\sum_{j \neq y_i} e^{\boldsymbol{R}_{G_i}^T \beta_j}\right) - \boldsymbol{R}_{G_i}^T \beta_{y_i}$.

According to Definition 3.5, $f(z) := \ln(1 + e^z)$ is a $(1, \ln 2, \ln 2)$-nice hinge function. Therefore, following Theorem C.1, if we sample subgraphs proportionally to the $\ell_1$ Lewis weights, we will obtain a $(1 \pm \epsilon)$-relative error coreset with probability at least $1 - \delta$, where $\delta > 0$ is a small constant. Specifically, if the sampling matrix $S \in \mathbb{R}^{m \times c}$ has each row chosen independently as the $i^{th}$ standard basis vector scaled by $1/p_i$ with probability $p_i/m$, then there exists a small $\epsilon > 0$ such that for any $\beta \in \mathbb{R}^{c \times C}$,

$$
\left| \sum_{i=1}^{m} [Sf(z)]_i - L(\beta) \right| \le \epsilon \cdot L(\beta).
\tag{22}
$$

$\square$

## D EXPERIMENTAL DETAILS

The detailed statistics for the datasets used for heterophilic graph active learning are shown in Table 4. We use effective GAL methods as baselines. Some methods are not selected since their code is not available (Song et al., 2023; Cui et al., 2022), or they focus on other settings, like noisy oracle (Zhang et al., 2021b; 2024). We briefly introduced the used baselines as follows:

Table 4: The statistics of used datasets.

| Dataset | #Nodes | #Edges | #Feature | #Class | $h$ |
|---|---|---|---|---|---|
| Roman-empire | 22,662 | 32,927 | 300 | 18 | 0.0469 |
| Amazon-ratings | 24,492 | 93,050 | 300 | 5 | 0.3804 |
| Tolokers | 11,758 | 519,000 | 10 | 2 | 0.5945 |
| Minesweeper | 10,000 | 39,402 | 7 | 2 | 0.6828 |
| Wisconsin | 251 | 499 | 1,703 | 5 | 0.1703 |
| Texas | 183 | 309 | 1,703 | 5 | 0.0615 |
| Snap-patents | 2,923,922 | 13,975,788 | 269 | 5 | 0.07 |

- Random: The naive random sampling that chooses nodes uniformly.

- Uncertainty (Settles & Craven, 2008): A GAL that chooses the nodes with maximum entropy on the predicted distribution.

- Density (Cai et al., 2017): A GAL that performs clustering on the embeddings of the nodes, and then chooses nodes with maximum density score.

- AGE (Cai et al., 2017): A GAL that selects nodes based on centrality, density, and uncertainty.

- ALG (Zhang et al., 2021a): A GAL that maximizes the effective reception field.

- FeatProp (Wu et al., 2019b): A GAL that first performs clustering on the propagated features, and then chooses the nodes closest to the cluster centers.

- GraphPart (Ma et al., 2023): A GAL that first splits the graph into disjoint partitions and then selects representative nodes within each partition.

## E CASE STUDY

We provide the case study in Figure 6 over selected nodes for different GAL methods on the Roman-empire dataset. We observe that KyN indeed selects more connected nodes than previous GAL baselines, which follows our principle of "know your neighbors".

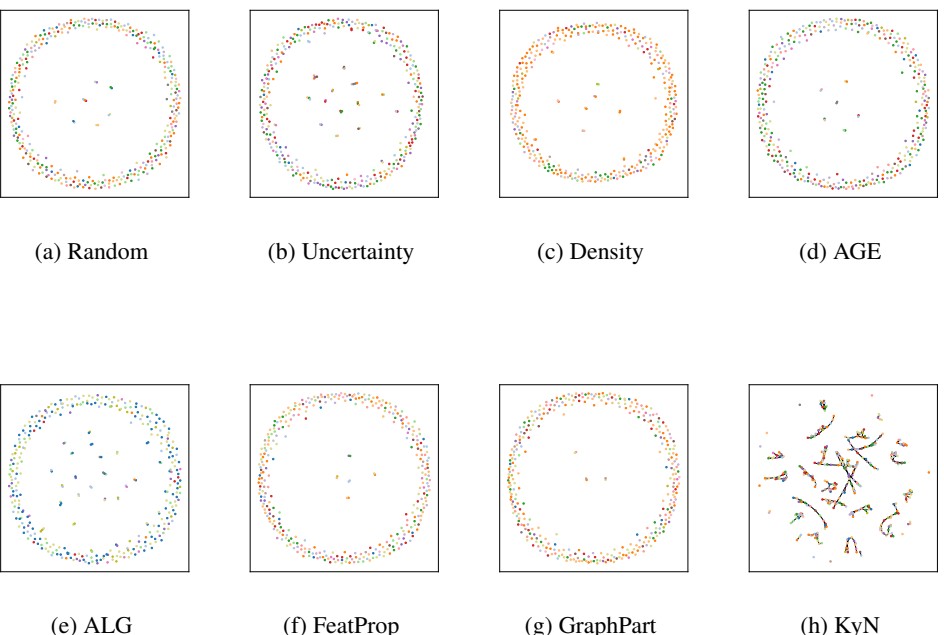

| (a) Random | (b) Uncertainty | (c) Density | (d) AGE |

| (e) ALG | (f) FeatProp | (g) GraphPart | (h) KyN |

Figure 6: The case study over selected nodes for different GAL methods on the Roman-empire dataset. The labeling budget is $20C$.

## F PSEUDOCODE

Algorithm 1 is the pseudocode of $\ell_1$ Lewis weights computation. The approximation has the time complexity of $\tilde{O}(\text{nnz}(\boldsymbol{M}) + d^\omega)$, where $d$ is the number of dimensions and $\omega \approx 2.37$ is the constant of fast matrix multiplication (Mai et al., 2021).

Algorithm 2 is the pseudocode of our KyN.

---

**Algorithm 1** $\ell_1$ Lewis weights computation. (Cohen & Peng, 2015)

---

**Require:** A representation matrix $M$, the approximation coefficient $\beta$, the iteration steps $T$.

  $w = \text{LewisIterate}(M, \beta, w)$
    **for** $i = 1 \ldots n$ **do**
      Let $\tilde{\tau}_i \approx_\beta \tau_i(W^{-1/2}M)$ be a $\beta$-approximation of the statistical leverage score of row $i$
    in $W^{-1/2}M$, where $W$ is the diagonal matrix of $w$.
      Set $\hat{w}_i \leftarrow (w_i \tilde{\tau}_i)^{1/2} \approx_{\beta^{1/2}} (m_i^T (M^T W^{-1} M)^{-1} m_i)^{1/2}$.
    **end for**
    **return** $\hat{w}$.
  $w = \text{ApproxLewisWeights}(M, \beta, T)$
    Initialize $w_i = 1$.
    **for** $t = 1 \ldots T$ **do**
      Set $w \leftarrow \text{LewisIterate}(M, \beta, w)$.
    **end for**
    **return** $w$.

---

**Algorithm 2** KyN

---

**Require:** A unlabeled graph $G$, the labeling budget $b$, the number of cluster $c$.

  Partition the graph into $c$ groups $V = \{V_1, \cdots, V_c\}$ with the METIS algorithm.
  Compute the subgraph representation $R$ with Eq. 8.
  Compute the subgraph $\ell_1$ Lewis weights $w = \text{ApproxLewisWeights}(R, \beta, T)$.
  Initialize count $= 0$ and the training set $V_{\text{train}} = \emptyset$.
  **while** count $< b$ **do**
    Sample a subgraph $V_i$ with the $\ell_1$ Lewis weights $w$.
    **if** count $+ |V_i| < b$ **then**
      Add all nodes in $V_i$ to $V_{\text{train}}$.
    **else**
      Add the central node of $|V_i|$ to $V_{\text{train}}$ and uniformly sample $b - \text{count} - 1$ nodes from $|V_i|$
  to $V_{\text{train}}$.
    **end if**
    count $= |V_{\text{train}}|$.
  **end while**
  **return** $V_{\text{train}}$.

---

## G  MORE NODE HOMOPHILY DISTRIBUTION PLOT

Figure 7 is the local node homophily distribution plot of different GAL-selected training sets and that of the ground truth on the Amazon-ratings dataset.

## H  DETAILED COMPONENT STUDIES

**Effectiveness of the importance sampling.** We use two datasets, Roman-empire and Tolokers, with a budget of $5C$ to test the effectiveness of the importance sampling. The results are presented in Table 5. We observe even without the full importance sampling, our model is still better than the naive random sampling. However, the performance degrades without the representative information.

Table 5: The ablation study of KyN to examine the effectiveness of the importance sampling. The labeling budget is $5C$.

|  | Roman-empire | Tolokers |
| --- | --- | --- |
| KyN | 44.8 | 71.0 |
| KyN w.o. Importance sampling | 43.7 | 68.5 |
| KyN w.o. Concatenation | 44.0 | 70.3 |

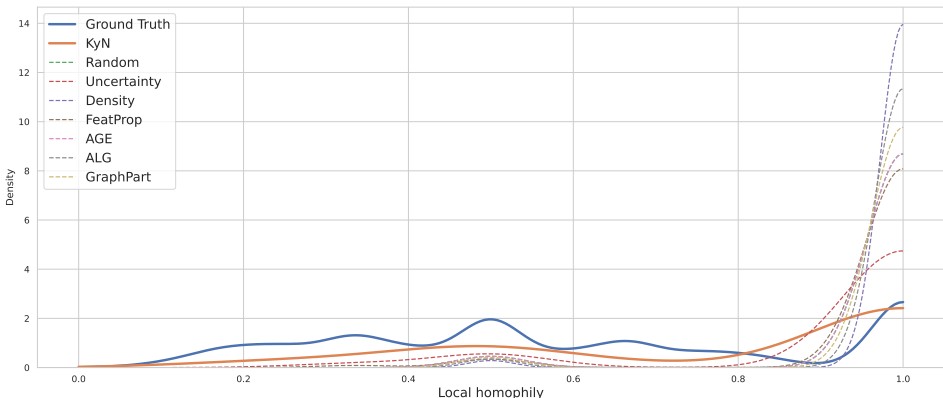

Figure 7: The local node homophily distribution plot of different GAL-selected training sets and that of the ground truth on the Amazon-ratings dataset. For a clear comparison, we also include our algorithm KyN. It is clear that KyN is the most similar to the ground truth distribution, and the only heterophilic one. We clip the distributions at 0 and 1. The labeling budget is $20C$.

**Why not "sample then select"?** We implement a simple "sample then select one-hop" method that first randomly select nodes and their one-hop neighbors. The results are presented in Table 6. We observe that the "sample then select $k$-hop" scheme is indeed suboptimal.

Table 6: The comparison between the two scheme, "sample then select $k$-hop" and "partition then sample". The labeling budget is $5C$.

|  | Roman-empire | Amazon-ratings | Tolokers |
|---|---|---|---|
| Sample then select one-hop | 43.4 | 30.5 | 66.7 |
| KyN | 44.8 | 31.2 | 71.0 |

**Different choice of graph partition methods.** We use METIS as our graph clustering algorithm as it is the de facto in the GNN realm (Chiang et al., 2019; Fey et al., 2021; Huang et al.). To justify our choice, we replace METIS with three algorithms, algebraic JC, variation neighborhoods, and affinity GS. The results are presented in Table 7. We observed that METIS performs the best, but the results of other graph clustering algorithms are also acceptable.

Table 7: The comparison between different graph partition methods. The labeling budget is $5C$.

|  | Roman-empire | Amazon-ratings | Tolokers |
|---|---|---|---|
| KyN+JC | 43.9 | 30.8 | 68.7 |
| KyN+VN | 44.2 | 30.8 | 69.2 |
| KyN+GS | 44.5 | 31.0 | 70.5 |
| KyN | 44.8 | 31.2 | 71.0 |

