# OpenReview forum: "Know Your Neighbors: Subgraph Importance Sampling for Heterophilic Graph Active Learning"
_ICLR.cc/2025/Conference — Submitted to ICLR 2025_

### Official Review · Reviewer_AM9W · 2024-10-17

**Soundness:** 3
**Presentation:** 3
**Contribution:** 2
**Rating:** 6
**Confidence:** 4

**Summary:**

This work points out the limitation of previous Graph Active Learning (GAL), which mainly focused on the homophilic graph. Consequently, they fail to outperform the naive random sampling algorithms, degrading the generalization ability of the GNNs. To solve this limitation, the author suggests a novel concept of "Know your neighbors" (KyN) with theoretical guarantee, which calculates $l_1$ Lewis weights and then samples the subgraph based on these probabilities. Extensive experiments on real-world datasets show the effectiveness of the proposed scheme.

**Strengths:**

**S1.** This work handles the limitation of GAL, which fail to perform well under heterophilic settings.

**S2.** The authors introduced some app appropriate examples, e.g., Figure 2 or 3, which helps to improve the readability of this paper.

**S3.** The proposed scheme (subgraph sampling with Lewis weight) has a fair theoretical guarantee.

**Weaknesses:**

**W1.** The KyN primarily targets heterogeneous graphs, but it employs GraphSAGE as a baseline. Fundamentally, GraphSAGE utilizes a mean aggregator (aggregation with positive edge weights lead to node convergence), which is less effective for heterogeneous edges, and therefore, the KyN may enhance performance under such conditions. The key point is that if the KyN can demonstrate its capabilities when integrated with heterogeneous GNN algorithms [2, 3]. Since the KyN introduces additional computational costs, if it fails to improve the quality of these baselines, it is questionable whether to apply the proposed GAL method in this community.
  * [1]: Beyond low-frequency information in graph convolutional networks, AAAI '21
  * [2]: Finding global homophily in graph neural networks when meeting heterophily, NeurIPS '22

Q1) Could you please run the experiments using the heterophilic GNN algorithms as above?


**W2.** The experiments could be further enhanced. For instance, in Table 1, the author should consider incorporating more baselines, such as GreedyET, and evaluate the node classification accuracy as demonstrated in [1]. Moreover, the paper neglects to utilize widely employed homophilic networks (e.g., Cora, Citeseer, and PubMed). As illustrated in Figure 2, the proposed KyN exhibits poor performance under high local homophily (1.0). It would be beneficial to investigate these values under homophilic networks.
  * [3] Cost-effective data labeling for graph neural networks, WWW '24

Q2) Could you use more baselines (e.g., GreedyET) and run the experiments under benchmark homophilic graphs (Cora, ...) as well?

**W3.** As illustrated in Figure 2, most nodes are concentrated around a homophily ratio of 0.5, from which the sampled subgraphs will be drawn, and KyN utilizes all nodes within these networks for training. In Theorem 3.4, the author proves that the sampling error converges to the coreset, which is acceptable. However, it remains unclear how the sampled subgraphs with low homophily (less than 0.5) contribute to enhancing the overall performance. Specifically, what happens if the homophily of most subgraphs is very low? According to [4,5], even low-homophily subgraphs can contribute to the node classification task by discovering patterns among neighboring nodes. Nevertheless, since KyN employs GraphSAGE, which involves positive message-passing, the neighboring nodes are smoothed during aggregation rather than being separated. Consequently, for KyN to improve classification accuracy, the sampled subgraphs with the same homophily ratio should ideally share the same class distribution, which may not be feasible.

  * [4] Is Heterophily A Real Nightmare For Graph Neural Networks To Do Node Classification?, arXiv '20
  * [5] Beyond homophily in graph neural networks: Current limitations and effective designs, NeurIPS '20

Q3) Could you provide a theoretical guarantee that the sampled subgraphs can still contribute to the overall performance despite having a low homophily ratio?

**Questions:**

Please refer to the weaknesses above

---

> ### Author Response · Authors · 2024-11-19
> **Response to Reviewer AM9W (Part 1)**
>
> Dear Reviewer AM9W,
>
> Thank you for your detailed review. Your suggestions are comprehensive and come with references, which is very helpful to us. We would like to address your questions/concerns below.
>
> > W1 & Q1: The KyN primarily targets heterogeneous graphs, but it employs GraphSAGE as a baseline. Fundamentally, GraphSAGE utilizes a mean aggregator (aggregation with positive edge weights lead to node convergence), which is less effective for heterogeneous edges, and therefore, the KyN may enhance performance under such conditions. The key point is that if the KyN can demonstrate its capabilities when integrated with heterogeneous GNN algorithms [2, 3]. Since the KyN introduces additional computational costs, if it fails to improve the quality of these baselines, it is questionable whether to apply the proposed GAL method in this community.
>
> This is indeed an important question, we understand your concern about the backbone, since we must eliminate the potential impact of inappropriate backbone. But we want to kindly point out that GraphSAGE *concatenates* the embedding of each target node and the average of its neighbors, that alows the so-called "negative aggregation". The "mean aggregator" is used only for the embedding of each node's neighbors. We are aware that the "ego-neighbor separation" is very crucial to heterophilic graphs[1,2]. That is precisely why we use GraphSAGE (to avoid the impact of inappropriate backbones) instead of GCN or GAT. We believe our rationale aligns with your considerations on this matter.
>
> We also agree that more heterophilic GNNs would make our paper better, so we have conducted more experiments based on your suggestions. The following is the result of two heterophilic GNNs on Roman-empire with the labeling budget of $20C$. We use FAGCN based on your suggestion and a very recent paper in ICML2024, M2M-GNN.
>
> |             | FAGCN          | M2M-GNN        |
> |-------------|----------------|----------------|
> | Random      | 52.0 $\pm$ 0.5 | 58.3 $\pm$ 1.3 |
> | Uncertainty | 47.7 $\pm$ 1.8 | 54.9 $\pm$ 1.0 |
> | Density     | 45.3 $\pm$ 1.1 | 51.5 $\pm$ 1.2 |
> | AGE         | 50.5 $\pm$ 1.9 | 55.7 $\pm$ 1.4 |
> | ALG         | 51.2 $\pm$ 1.3 | 56.3 $\pm$ 1.2 |
> | FeatProp    | 51.7 $\pm$ 0.9 | 57.0 $\pm$ 0.8 |
> | GraphPart   | 51.6 $\pm$ 1.2 | 56.1 $\pm$ 0.8 |
> | KyN(Ours)   | 53.5 $\pm$ 1.4 | 59.2 $\pm$ 1.0 |
>
> We observe that on these two heterophilic GNNs, our KyN also selects valuable training set, which further supports our argument.
>
> [1] A critical look at the evaluation of GNNs under heterophily: Are we really making progress? ICLR2023.
>
> [2] Beyond homophily in graph neural networks: Current limitations and effective designs. NeurIPS2020.
>
> [3] Beyond low-frequency information in graph convolutional networks. AAAI2022.
>
> [4] Sign is Not a Remedy: Multiset-to-Multiset Message Passing for Learning on Heterophilic Graphs. ICML2024.
>
> > W2 & Q2. The experiments could be further enhanced. For instance, in Table 1, the author should consider incorporating more baselines, such as GreedyET, and evaluate the node classification accuracy as demonstrated in [1]. Moreover, the paper neglects to utilize widely employed homophilic networks (e.g., Cora, Citeseer, and PubMed). As illustrated in Figure 2, the proposed KyN exhibits poor performance under high local homophily (1.0). It would be beneficial to investigate these values under homophilic networks.
>
> Thank you for your question. We first want to kindly point out that Figure 2 is not a performance figure. It shows the distribution of local homophily. The density of KyN is low near 1 because it correctly reflects the real distribution, as the density of ground truth (the blue line) is also low near 1.
>
> In our original submission, we do not include GreedyET as baseline because we fail to reproduce its performance. We will continue to try to run its code, but we are not sure it can be done at the rebuttal stage. For homophilic datasets, we do not think our method will have any particular advantages because we are mainly addressing the problem of misestimation of homophily by traditional GALs on heterophilic graphs.

---

> ### Author Response · Authors · 2024-11-19
> **Response to Reviewer AM9W (Part 2)**
>
> > W3 & Q3. As illustrated in Figure 2, most nodes are concentrated around a homophily ratio of 0.5, from which the sampled subgraphs will be drawn, and KyN utilizes all nodes within these networks for training. In Theorem 3.4, the author proves that the sampling error converges to the coreset, which is acceptable. However, it remains unclear how the sampled subgraphs with low homophily (less than 0.5) contribute to enhancing the overall performance. Specifically, what happens if the homophily of most subgraphs is very low? According to [4,5], even low-homophily subgraphs can contribute to the node classification task by discovering patterns among neighboring nodes. Nevertheless, since KyN employs GraphSAGE, which involves positive message-passing, the neighboring nodes are smoothed during aggregation rather than being separated. Consequently, for KyN to improve classification accuracy, the sampled subgraphs with the same homophily ratio should ideally share the same class distribution, which may not be feasible.
>
> Thank you for your questions. Based on your suggestion, we have proved **two more theorems**. We have uploaded the new version. The first new theorem says that a correct (whether low or high) local homophily is necessary for a high accuracy. We believe this will answer your insightful question of "Could you provide a theoretical guarantee that the sampled subgraphs can still contribute to the overall performance despite having a low homophily ratio?" The second theorem is about the benefits of our "know your neighbors".
>
> As mentioned earlier in our response to your insightful Q1, for the GraphSAGE backbone, it has the ability to do "negative message-passing" since it separates the ego- and neighbor-embedding.
>
> Thanks again for the detailed review. Your suggestions are very helpful to us and we will continue to improve this paper.

---

> ### Author Response · Authors · 2024-11-25
> **Looking forward to your reply**
>
> Dear Reviewer AM9W,
>
> As we are days away from the closure of the discussion phase, we eagerly await your feedback on the revisions made. We understand that you may have put in a lot of effort during the discussion phase, and we truly appreciate your hard work. As reviewers for the other papers, we responded to the authors promptly. We would greatly appreciate it if you could also take a look at the changes we made.
>
> Your suggestions have been extremely helpful to us. Based on your feedback, we conducted the GAL experiment with **two additional backbones**, **proved two new theorems**. We believe that the current version is a significant improvement over the initial submission, and we sincerely hope to earn your approval.

---

> > ### Comment · Reviewer_AM9W · 2024-11-26
> > **Thanks for the rebuttal**
> >
> > Dear authors,
> >
> > Sorry for the late reply. As most of my concerns have been addressed, I have increased my score to 6.

---

> > > ### Author Response · Authors · 2024-11-27
> > > **Thank you**
> > >
> > > Dear Reviewer AM9W,
> > >
> > > We sincerely thank you for your reply! You really have reignited our hope. We will upload the final version soon and you can take a look at it if you have some spare time.
> > >
> > > We wish you good luck in the extended discussion period!
> > >
> > > Best regards,
> > >
> > > Authors

---

### Official Review · Reviewer_LKPL · 2024-10-28

**Soundness:** 3
**Presentation:** 1
**Contribution:** 2
**Rating:** 5
**Confidence:** 3

**Summary:**

The KYN model (Know Your Neighbors) is designed to address the limitations of traditional Graph Neural Networks (GNNs) in handling heterophilic graphs (graphs where nodes with the same labels are less likely to be connected). While existing Graph Active Learning (GAL) methods work well on homophilic graphs (where similar nodes are connected), they perform poorly on heterophilic graphs, often even worse than random sampling. The KYN model introduces a novel approach to solve this problem.And through theoretical analysis and experiments to demonstrate the effectiveness and superiority of the model.

**Strengths:**

1. The "Know Your Neighbors" approach is innovative and ensures that nodes and their neighbors are labeled together, providing GNNs with the correct local homophily distribution. This method may significantly enhance the learning performance on heterophilic graphs
2. Subgraph importance sampling: By leveraging ℓ1 Lewis weights (which offers theoretical guarantees for the quality of the sampled subgraphs.)for subgraph sampling, my understanding is that KYN selects subgraphs that are most informative, reducing the risk of overfitting to uninformative nodes and ensuring that the labeled nodes provide useful information for training.
3. The theoretical guarantee proves the rationality and effectiveness of sampling strategy based on ℓ1 Lewis weight in graph learning; analyzes the influence of subgraph division and subgraph embedding strategy on node classification task in heterograph; and explains the advantages of separating self-embedding and neighbor embedding for model processing of heterograph.

**Weaknesses:**

1. The quality of subgraph division depends on the clustering algorithm: KYN uses a graph clustering algorithm (such as METIS) to divide subgraphs, and the performance and effect of these clustering algorithms directly affect the representation of subgraph and subsequent model training. If the clustering results are not of high quality, it may lead to the subgraph internal structure is not compact, which will affect the overall model performance. I think it is necessary to further discuss the performance of the clustering algorithm used in the partition subgraphs.
2. Lack of clear description of the algorithm steps: When introducing the execution steps of KYN model, there may be some operational details not detailed, especially some key processes (such as specific implementation of subgraph division, calculation of Lewis weight, etc.) without clear pseudo code or algorithm block diagram. This may leave the reader for difficulty in actually reproducing or understanding the algorithm.
3. Insufficient explanation of the dataset: When introducing the dataset used for the experiment, the article does not fully explain the structure, characteristics, and the degree of graph heterogeneity of each dataset. This information is essential for understanding the experimental results as well as the applicability of KYN.
In addition, to further verify the robustness of the KYN model, experiments may be conducted on more graph types, such as:Large-scale graphs、Dynamic graphs、graphs with different connection densities.
4. A small detail problem, when Texas data set 20C, Uncertainty (the second row in the table) corresponds to the best result, not KyN

**Questions:**

See the above weakness

---

> ### Author Response · Authors · 2024-11-19
> **Response to Reviewer LKPL**
>
> Dear Reviewer LKPL,
>
> We thank you for the insightful review. Your recognition of our novelty and impact is very important to us. We would like to address your questions/concerns below.
>
> > W1: The quality of subgraph division depends on the clustering algorithm: KYN uses a graph clustering algorithm (such as METIS) to divide subgraphs, and the performance and effect of these clustering algorithms directly affect the representation of subgraph and subsequent model training. If the clustering results are not of high quality, it may lead to the subgraph internal structure is not compact, which will affect the overall model performance. I think it is necessary to further discuss the performance of the clustering algorithm used in the partition subgraphs.
>
> We thank you for this insightful question. We use METIS as our graph clustering algorithm as it is the de facto in the GNN realm. It was used in many excellent works, e.g., [1-3]. We agree that it would be better to discuss the performance of the clustering algorithm, so we have conducted additional experiments based on your suggestions. We replace METIS with three algorithms, algebraic JC, variation neighborhoods and affinity GS. We choose three datasets and set the budget to $5C$ due to the limited time in the discussion phase. The results are as follow.
>
> |        | Roman-empire | Amazon-ratings | Tolokers |
> |--------|--------------|----------------|----------|
> | KyN+JC | 43.9         | 30.8           | 68.7     |
> | KyN+VN | 44.2         | 30.8           | 69.2     |
> | KyN+GS | 44.5         | 31.0           | 70.5     |
> | KyN    | 44.8         | 31.2           | 71.0     |
>
> We observed that METIS performs the best, but the results of other graph clustering algorithms are also acceptable.
>
> [1]Cluster-gcn: An efficient algorithm for training deep and large graph convolutional networks. KDD2019.
>
> [2]Gnnautoscale: Scalable and expressive graph neural networks via historical embeddings. ICML2021.
>
> [3]Cluster-wise Graph Transformer with Dual-granularity Kernelized Attention. NeurIPS2024.
>
> > W2: Lack of clear description of the algorithm steps: When introducing the execution steps of KYN model, there may be some operational details not detailed, especially some key processes (such as specific implementation of subgraph division, calculation of Lewis weight, etc.) without clear pseudo code or algorithm block diagram. This may leave the reader for difficulty in actually reproducing or understanding the algorithm.
>
> We sincerely thank you for this suggestions, as it greatly help us to imporve the readability of our paper. We have added these pseudocode to the appendix.
>
> > W3: Insufficient explanation of the dataset: When introducing the dataset used for the experiment, the article does not fully explain the structure, characteristics, and the degree of graph heterogeneity of each dataset. This information is essential for understanding the experimental results as well as the applicability of KYN. In addition, to further verify the robustness of the KYN model, experiments may be conducted on more graph types, such as:Large-scale graphs, Dynamic graphs, graphs with different connection densities.
>
> Thank you for your question, the statistics of dataset are included in the original submission. It is in Appendix B. We agree more dataset would be better, so we have conducted more experiments on a large-scale graph based on your suggestion. We use Snap-patents. It has 2,923,922 nodes and 13,975,788 edges. We set the budget to $5C$ and compare KyN with previous GALs.
>
> |             | Accuracy | Time (second) |
> |-------------|----------|------|
> | Random      | 32.9     | 0.06 |
> | Uncertainty | 25.5     | 0.45 |
> | Density     | 25.1     | 752  |
> | AGE         | -        | OOT  |
> | ALG         | -        | OOT  |
> | FeatProp    | -        | OOT  |
> | GraphPart   | -        | OOT  |
> | KyN(Ours)   | 33.7     | 651  |
>
> Due to the limited time during the discussion phase, we record GALs exceeding 30 minutes as OOT (out of time). We observe KyN achieve SOTA performance on this large heterophilic graph, and the runtime is reasonable. This experiments on large graph shows the efficiency of our KyN.
>
> > W4: A small detail problem, when Texas data set 20C, Uncertainty (the second row in the table) corresponds to the best result, not KyN
>
> Thank you for pointing that out. We have fixed the typo.
>
> We would like to thank you once again for reviewing our paper. Your suggestions have been incredibly helpful! We spent a significant amount of time conducting the experiments you proposed, and we would greatly appreciate any further feedback you could provide. We look forward to your continued support.

---

> ### Author Response · Authors · 2024-11-25
> **Looking forward to your reply**
>
> Dear Reviewer LKPL,
>
> As we are days away from the closure of the discussion phase, we eagerly await your feedback on the revisions made. We understand that you may have put in a lot of effort during the discussion phase, and we truly appreciate your hard work. As reviewers for the other papers, we responded to the authors promptly. We would greatly appreciate it if you could also take a look at the changes we made.
>
> Your suggestions have been extremely helpful to us. Based on your feedback, we conducted **two additional experiments**, and improved our presentation, including **adding pseudocode**. We believe that the current version is a significant improvement over the initial submission, and we sincerely hope to earn your approval.

---

> > ### Comment · Reviewer_LKPL · 2024-11-25
> >
> > Thank you for your response. I greatly appreciate and commend the effort the authors have invested during the rebuttal phase. However, after carefully considering the current completeness of the manuscript, the feedback from other reviewers, and the rebuttal content, I am inclined to maintain the current score. I encourage the authors to further refine their work and wish them success in future submissions.

---

> > > ### Author Response · Authors · 2024-11-26
> > > **Thank you**
> > >
> > > Dear Reviewer LKPL,
> > >
> > > Thank you for your encouragement and we respect your decision. We will continue to improve our paper based on your suggestions, even if we may not get it accepted this time. We will also incorporate these precious discussions in the manuscript, a final version will be uploaded within one or two days and you can take a look at it if you have some spare time.
> > >
> > > We wish you good luck in the extended discussion period!
> > >
> > > Best regards,
> > >
> > > Authors

---

### Official Review · Reviewer_kN83 · 2024-11-01

**Soundness:** 1
**Presentation:** 2
**Contribution:** 1
**Rating:** 1
**Confidence:** 5

**Summary:**

The paper tried to bring attention to graph active learning for heterophilic graphs by making several claims that are not prudent as pointed out in the weaknesses. To conclude, the paper lacks severely on the experimentation and also on the way it has been conducted. Moreover the absolute improvements in results are just marginal, calculating percentages just inflates these values. Most importantly, the claims made in the paper are unjustified.

**Strengths:**

1.) The paper presents a technique for Graph Active Learning (GAL) where it samples subgraphs based on Lewis weights sampling by using them as probabilities and name it as “Know your Neighbors”.

2.) The paper claims to “investigate a new research problem, heterophilic graph active learning”. Although it is not correct as discussed below.

**Weaknesses:**

1.) The basic premise of the paper, i.e., the claim “previous GAL methods fail to outperform the naive random sampling on heterophilic graphs” is not substantial. Out the 6 reported heterophilic datasets, only for Roman-empire dataset, random sampling is better than the previous GAL methods, and for Minesweeper dataset, it is almost similar. So, this blanket statement should not have been made.

2.) It is not just the homophily that influences the performance of any GNN on a given graph but many other network properties and also the data split ratio and the random seed. Moreover, the definition of homophily itself is not just limited to node homophily and the edge homophily which the authors have referred to as local homophily and global homophily respectively, but many other homophily measures like improved homophily [1], adjusted homophily [2], effective homophily [3] and aggregated homophily [4]. When you take the average of homophily for all the nodes in the graph it becomes a global measure, i.e., node homophily, so referring to it as local homophily is not appropriate. The authors are suggested to incorporate these additional measures to have a better clarity.

3.) The authors have just shown the homophily distribution (just the local homophily) plot in Fig.2 only for one dataset and inferred and made a claim that “previous GAL methods provide wrong, even opposite, information in the first place”. If we are to go by this claim, the results of previous GAL methods would yield gibberish which it is clearly not. I strongly recommend the authors to moderate their writing regarding the performance of previous GAL methods and provide more nuanced analysis instead of making harsh claims. The authors should provide homophily distribution plots for all other datasets as well to make their point.

4.) Also, for homophily calculations, you do not use the node itself resulting in a self loop. It is wrong, c.f. 183-189 that also forms the base of deriving the proposed technique. Moreover, it raises a serious question on the validity of the homophily distribution plot in Fig.2 that for the majority of the nodes selected through GAL which are isolated, value of local homophily of 1 is assumed.

5.) The paper talks about graph partitioning as the primary step and chooses METIS algorithm for the rest of the work. Why? It is a building block of the proposed technique, atleast a few different graph clustering/partitioning algorithms should have been discussed and experimented with.

6.) Similarly, for graph aggregation, GraphSAGE has been used which is again a rudimentary method just like the above chosen METIS.

7.) The statement “We want to point out that GAL methods can be used with all previously mentioned GNNs that designed for heterophilic graphs. We omit such combinations without loss of generality.” is troublesome. Solid experimentation needs to be performed to arrive at this claim and I suggest authors to provide extensive empirical evidence to support this claim.

8.) In Table1, the values for Average improvement correspond to which of the mentioned baselines? Is it with respect to the second best result?

9.) By the very principle of the proposed technique, it should work perfectly for homophilic graphs. Why is there no such discussion and the corresponding results?

10.) The two statements “We observe that KyN is robust to the choice of c.” and “This is normal since we do not tune c with the final accuracy to avoid data leakage.” are problematic. For many of the datasets, the accuracy fluctuates significantly by almost 10 absolute points, so how is KyN robust to the choice of c? Ofcourse, one does not tune hyperparameter with the final accuracy to avoid data leakage on the test set but it has to be done on the validation set. I recommend the authors to clarify their hyperparameter tuning process, particularly regarding the use of a validation set.

11.) The statement “In practice, we recommend choosing c around |V |/C , where C is the number of classes.”, what is the basis of the given statement?

12.) There are slight grammatical mistakes like “theoratical” in line 266 etc.

###########
References:

[1] Large scale learning on non-homophilous graphs: New benchmarks and strong simple methods. Advances in Neural Information Processing Systems, 34:20887–20902, 2021.

[2] Characterizing graph datasets for node classification: Beyond homophily-heterophily dichotomy. arXiv preprint arXiv:2209.06177, 2022.

[3] Edge directionality improves learning on heterophilic graphs. Proceedings of the Second Learning on Graphs Conference, volume 231 of Proceedings of Machine Learning Research, pp. 25:1–25:27. PMLR, 27–30 Nov 2024. URL https://proceedings.mlr.press/v231/rossi24a.html.

[4] Revisiting heterophily for graph neural networks. Advances in neural information processing systems, 35:1362–1375, 2022.

**Questions:**

Please see above in the weaknesses section.

---

> ### Author Response · Authors · 2024-11-19
> **Response to Reviewer kN83 (Part 1)**
>
> **We first want to clarify that our paper does not mean to attack previous GALs or disrespect them.** It seems there has been a significant misunderstanding regarding our intentions. We think previous GALs were great, but they were not designed for heterophilic graphs. Our intention was simply to highlight the potential harm heterophily could cause to GALs and to offer a solution. Below are our responses to your questions.
>
> > W1. The basic premise of the paper, i.e., the claim “previous GAL methods fail to outperform the naive random sampling on heterophilic graphs” is not substantial. Out the 6 reported heterophilic datasets, only for Roman-empire dataset, random sampling is better than the previous GAL methods, and for Minesweeper dataset, it is almost similar. So, this blanket statement should not have been made.
>
> Given the simplicity of random sampling, it is concerning enough that it outperforms previous GALs on some datasets. And we did not claim random sampling outperforms them in all datasets.
>
> > W2. It is not just the homophily that influences the performance of any GNN on a given graph but many other network properties and also the data split ratio and the random seed. Moreover, the definition of homophily itself is not just limited to node homophily and the edge homophily which the authors have referred to as local homophily and global homophily respectively, but many other homophily measures like improved homophily [1], adjusted homophily [2], effective homophily [3] and aggregated homophily [4]. When you take the average of homophily for all the nodes in the graph it becomes a global measure, i.e., node homophily, so referring to it as local homophily is not appropriate. The authors are suggested to incorporate these additional measures to have a better clarity.
>
> Local homophily is a term commonly used by the community (e.g.,[1,2]). We think it is not inappropriate to refer to it by its original name. In this paper, we use local homophily to perform a novel analysis on the GAL-selected training set and that gives us novel insights. We believe that additional homophily measures are not particularly relevant to the focus of our work.
>
> [1]On performance discrepancies across local homophily levels in graph neural networks. LOG2023.
>
> [2]Demystifying structural disparity in graph neural networks: Can one size fit all? NeurIPS2024.
>
> > W3: The authors have just shown the homophily distribution (just the local homophily) plot in Fig.2 only for one dataset and inferred and made a claim that “previous GAL methods provide wrong, even opposite, information in the first place”. If we are to go by this claim, the results of previous GAL methods would yield gibberish which it is clearly not. I strongly recommend the authors to moderate their writing regarding the performance of previous GAL methods and provide more nuanced analysis instead of making harsh claims. The authors should provide homophily distribution plots for all other datasets as well to make their point.
>
> We thank you for your request to add more homophily distribution plots. We have added an additional plot in the appendix.
>
> About your statement of "If we are to go by this claim, the results of previous GAL methods would yield gibberish". We beg to differ. GNNs will receive incorrect homophily distribution, but still the correct labels, so the performance will naturally be better than random guessing.
>
> > W4: Also, for homophily calculations, you do not use the node itself resulting in a self loop. It is wrong, c.f. 183-189 that also forms the base of deriving the proposed technique. Moreover, it raises a serious question on the validity of the homophily distribution plot in Fig.2 that for the majority of the nodes selected through GAL which are isolated, value of local homophily of 1 is assumed.
>
> We beg to differ. We use the self-loop and we think it is very reasonable. (1) GNNs do not discard each node's own feature, the embedding of each node is the most important feature to itself. (2) Without the self-loop, it would be impossible to compute the local homophily of isolated nodes, as this would result in an undefined expression ($\frac{0}{0}$).
>
> About your statement of "Moreover, it raises a serious question on the validity of the homophily distribution plot in Fig.2 that for the majority of the nodes selected through GAL which are isolated, value of local homophily of 1 is assumed". **We have already explained the validity in the original submission. It is in line 157-161.** And "the majority of the nodes selected through GAL which are isolated" is exactly why we propose the principle of KyN. We believe there is no confusion regarding this matter.

---

> ### Author Response · Authors · 2024-11-19
> **Response to Reviewer kN83 (Part 2)**
>
> > W5: The paper talks about graph partitioning as the primary step and chooses METIS algorithm for the rest of the work. Why? It is a building block of the proposed technique, atleast a few different graph clustering/partitioning algorithms should have been discussed and experimented with.
>
> Thanks for the question. We use METIS as our graph clustering algorithm as it is the de facto in the GNN realm. It was used in many excellent works, e.g., [1-3]. We have conducted additional experiments. We replace METIS with three algorithms, algebraic JC, variation neighborhoods and affinity GS. We choose three datasets and set the budget to $5C$ due to the limited time in the discussion phase. The results are as follow.
>
> |        | Roman-empire | Amazon-ratings | Tolokers |
> |--------|--------------|----------------|----------|
> | KyN+JC | 43.9         | 30.8           | 68.7     |
> | KyN+VN | 44.2         | 30.8           | 69.2     |
> | KyN+GS | 44.5         | 31.0           | 70.5     |
> | KyN    | 44.8         | 31.2           | 71.0     |
>
> We observed that METIS performs the best, but the results of other graph clustering algorithms are also competitive.
>
> [1]Cluster-gcn: An efficient algorithm for training deep and large graph convolutional networks. KDD2019.
>
> [2]Gnnautoscale: Scalable and expressive graph neural networks via historical embeddings. ICML2021.
>
> [3]Cluster-wise Graph Transformer with Dual-granularity Kernelized Attention. NeurIPS2024.
>
> > W6: Similarly, for graph aggregation, GraphSAGE has been used which is again a rudimentary method just like the above chosen METIS.
>
> We beg to differ. **We have clearly stated why we use GraphSAGE in section 5.1 of the original submission.**
>
> > W7: The statement “We want to point out that GAL methods can be used with all previously mentioned GNNs that designed for heterophilic graphs. We omit such combinations without loss of generality.” is troublesome. Solid experimentation needs to be performed to arrive at this claim and I suggest authors to provide extensive empirical evidence to support this claim.
>
> Thanks for this question. The following is the result of two heterophilic GNNs, FAGCN [1] and M2M-GNN [2] on Roman-empire with the labeling budget of $20C$.
>
> |             | FAGCN          | M2M-GNN        |
> |-------------|----------------|----------------|
> | Random      | 52.0 $\pm$ 0.5 | 58.3 $\pm$ 1.3 |
> | Uncertainty | 47.7 $\pm$ 1.8 | 54.9 $\pm$ 1.0 |
> | Density     | 45.3 $\pm$ 1.1 | 51.5 $\pm$ 1.2 |
> | AGE         | 50.5 $\pm$ 1.9 | 55.7 $\pm$ 1.4 |
> | ALG         | 51.2 $\pm$ 1.3 | 56.3 $\pm$ 1.2 |
> | FeatProp    | 51.7 $\pm$ 0.9 | 57.0 $\pm$ 0.8 |
> | GraphPart   | 51.6 $\pm$ 1.2 | 56.1 $\pm$ 0.8 |
> | KyN(Ours)   | 53.5 $\pm$ 1.4 | 59.2 $\pm$ 1.0 |
>
> We observe that on these two heterophilic GNNs, our KyN also selects valuable training set, which further supports our argument.
>
> [1] Beyond low-frequency information in graph convolutional networks. AAAI2022.
>
> [2] Sign is Not a Remedy: Multiset-to-Multiset Message Passing for Learning on Heterophilic Graphs. ICML2024.
>
> > W8: In Table1, the values for Average improvement correspond to which of the mentioned baselines? Is it with respect to the second best result?
>
> It is w.r.t all previous GALs.
>
> > W9: By the very principle of the proposed technique, it should work perfectly for homophilic graphs. Why is there no such discussion and the corresponding results?
>
> This paper studies the impact of **heterophily** on GALs. We did not claim it should work perfectly for homophilic graphs.
>
> > W10: The two statements “We observe that KyN is robust to the choice of c.” and “This is normal since we do not tune c with the final accuracy to avoid data leakage.” are problematic. For many of the datasets, the accuracy fluctuates significantly by almost 10 absolute points, so how is KyN robust to the choice of c? Ofcourse, one does not tune hyperparameter with the final accuracy to avoid data leakage on the test set but it has to be done on the validation set. I recommend the authors to clarify their hyperparameter tuning process, particularly regarding the use of a validation set.
>
> Subtracting the minimum from the maximum is not the right way to verify robustness. You can always find a hyperparameter that makes a model bad. When we say KyN is robust to the choice of c, we mean that reasonable choices can yield a satisfactory performance.
>
> In the GAL community, a validation set is usually unavailable in reality. Please refer to the abstract of [1]. That is why we recommend a hyperparameter choice.
>
> [1]Partition-Based Active Learning for Graph Neural Networks. TMLR2023.
>
> > W11: The statement “In practice, we recommend choosing c around |V |/C , where C is the number of classes.”, what is the basis of the given statement?
>
> This is to avoid having a giant connect component without making the algorithm take too long to run.
>
> >W12: There are slight grammatical mistakes like “theoratical” in line 266 etc.
>
> Thank you, we have fixed that.

---

> ### Author Response · Authors · 2024-11-25
> **Looking forward to your reply**
>
> Dear Reviewer kN83,
>
> As the discussion phase closes in two days, we sincerely hope to clear up any misunderstanding with you. Our paper in no way intends to criticize previous GAL works, and we have great respect and admiration for the authors of earlier works. We believe the misunderstanding may have arisen due to differences in cultural background and language conventions. If you still have any doubts about our intentions, please let us know what we can do to resolve the issue.

---

> > ### Comment · Reviewer_kN83 · 2024-11-29
> >
> > Dear Authors,
> >
> > I appreciate the structure of your summarised response at the top of the entire discussion thread. I also appreciate you for the long rebuttal and your response to W5, W7 and W12. However, I found the all other responses as evasive and failed to resolve my grave concerns. Hence, I cannot increase my rating.
> >
> > Thanks.

---

### Official Review · Reviewer_L9AD · 2024-11-03

**Soundness:** 2
**Presentation:** 2
**Contribution:** 2
**Rating:** 5
**Confidence:** 5

**Summary:**

This paper addresses graph active learning (GAL) on heterophilic graphs, a previously unexplored perspective. The authors argue that existing GAL methods underperform on heterophilic graphs due to their inability to preserve ground-truth local homophily distributions in the selected training set. To address this limitation, they propose selecting nodes along with their neighbors as the training set to better capture local homophily distribution. The proposed method, KyN, first clusters the input graph using METIS and identifies Jordan centers for each cluster. Cluster-level features are then constructed by concatenating center features with aggregated neighbor features. Clusters are subsequently sampled and labeled according to $\ell_1$ Lewis weights. Experimental results demonstrate the effectiveness of KyN on real-world homophilic graphs.

**Strengths:**

**S1**: The paper addresses an important gap in GAL research by focusing on homophilic graphs.

**S2**: The authors provide a novel perspective on why existing GAL methods fail on homophilic graphs, attributing it to inconsistencies in local homophily distribution estimation.

**S3**: The proposed method offers a simple, efficient solution by preserving ground-truth homophily degree through neighbor-aware labeling, demonstrating effectiveness across six real-world datasets.

**Weaknesses:**

**W1**: The central argument in this paper regarding the limitations of existing Graph Active Learning (GAL) methods lacks comprehensive empirical validation. The experimental methodology omits crucial details, particularly regarding GNN backbones and configurations. To establish a more convincing foundation, the analysis should incorporate results from both heterophilic GNN and MLP backbones. This expansion would provide stronger evidence for the claimed deficiencies in current approaches and demonstrate the broad applicability of the proposed solution across different GNN backbones.

**W2**: The key insight behind the proposed method requires further elaboration. While the authors claim that labeling subgraphs better captures local homophily distribution, the mechanism by which this information enhances GNN performance remains unclear. The manuscript should explicitly address several key questions. How does the degree of local homophily influence GNN behavior? What specific advantages does this approach offer for fundamental homophily and heterophily GNNs? A detailed theoretical analysis would strengthen the paper's contribution and provide valuable insights for future research directions.

**W3**: The experimental evaluation requires substantial enhancement in multiple dimensions:
- Comprehensive evaluation across diverse backbone architectures, specifically including heterophily GNN and MLP variants.
- Validation on large-scale datasets to substantiate efficiency claims and demonstrate practical applicability.
- Detailed ablation studies to isolate the impact of individual components.

**W4**: The presentation requires significant improvement. The core principle of "know your neighbor" lacks precise definition and theoretical grounding. The relationship between local homophily distribution and active learning effectiveness requires clearer articulation. A substantial revision would enhance readability and impact.

Overall, although heterophilic GAL presents an interesting research direction, the current manuscript has significant limitations in motivation evidence (W1), key insight (W2), experimental validation (W3), and presentation (W4).

**Questions:**

**Q1**: The experimental setup in Table 1 requires clarification regarding the feature combination mechanism in the backbone GNN. How are neighbor-aggregated features combined with target node features? Does the choice of combination method impact performance significantly?

**Q2**: The ablation study section requires expansion to validate the contribution of key components. How does the method perform when $\ell_1$ loss weights are excluded, such as in the case of randomly sampling a subgraph? What impact does the removal of concatenation operations from Equation (6) have on the model performance?

**Q3**: The experimental analysis in Figure 1 should be expanded, for example, by extending the labeling budget range to start from 2c to align with the standard GAL setting.

**Q4**: What is the performance comparison with a *sample-then-$k$-hop* approach? The terminology also needs revision (*sample* as verb vs. *k-hop* as noun).

**Q5**: How are node selections made when selected subgraphs exceed the budget? For example, the current budget is 2 and the selected subgraph contains 5 nodes, what selection criteria are applied?

**Q6**: Please supplement the $\ell_1$ Lewis weights computation algorithm and time complexity analysis.

---

> ### Author Response · Authors · 2024-11-19
> **Response to Reviewer L9AD (Part 1)**
>
> Dear Reviewer L9AD,
>
> We sincerely thank you for your detailed review, your suggestions are very valuable to us. We appreciate your recognition of our research questions, and we strive to improve our paper based on your suggestions. Here are the improvements we made and responses to your questions.
>
> > W1 & W3(1): The central argument in this paper regarding the limitations of existing Graph Active Learning (GAL) methods lacks comprehensive empirical validation. The experimental methodology omits crucial details, particularly regarding GNN backbones and configurations. To establish a more convincing foundation, the analysis should incorporate results from both heterophilic GNN and MLP backbones. This expansion would provide stronger evidence for the claimed deficiencies in current approaches and demonstrate the broad applicability of the proposed solution across different GNN backbones.
>
> Thanks for this insightful question. Based on your suggestions, we have conducted more experiments using two heterophilic GNNs, FAGCN [1] and M2M-GNN [2]. FAGCN is suggested by other reviewers and we pick M2M-GNN since it is a recent paper in ICML2024. We have also included MLP as a backbone. The following are the experimental results. The experiments are conducted on the Roman-empire dataset with $20C$ budgets.
>
> |             | FAGCN          | M2M-GNN        | MLP            |
> |-------------|----------------|----------------|----------------|
> | Random      | 52.0 $\pm$ 0.5 | 58.3 $\pm$ 1.3 | 53.1 $\pm$ 0.7 |
> | Uncertainty | 47.7 $\pm$ 1.8 | 54.9 $\pm$ 1.0 | 48.8 $\pm$ 2.6 |
> | Density     | 45.3 $\pm$ 1.1 | 51.5 $\pm$ 1.2 | 48.3 $\pm$ 2.3 |
> | AGE         | 50.5 $\pm$ 1.9 | 55.7 $\pm$ 1.4 | 53.6 $\pm$ 0.8 |
> | ALG         | 51.2 $\pm$ 1.3 | 56.3 $\pm$ 1.2 | 50.4 $\pm$ 1.7 |
> | FeatProp    | 51.7 $\pm$ 0.9 | 57.0 $\pm$ 0.8 | 52.3 $\pm$ 1.2 |
> | GraphPart   | 51.6 $\pm$ 1.2 | 56.1 $\pm$ 0.8 | 50.7 $\pm$ 1.3 |
> | KyN(Ours)   | 53.5 $\pm$ 1.4 | 59.2 $\pm$ 1.0 | 53.4 $\pm$ 0.9 |
>
> From the above results, we observe that the improvements are still valid on these two heterophilic GNNs. From a high-level perspective, this is because even heterophilic GNNs have the *ability* of learning heterophilic patterns, they cannot do so if the GAL-selected training sets fail to provide the heterophilic information. For MLP, most GALs perform poorly since they all consider graph structure when selecting nodes, which is redundant with a MLP backbone. We notice that AGE performs best among all GALs, this is reasonable since it has hyperparameters that control the effect of graph structure.
>
> [1] Beyond low-frequency information in graph convolutional networks. AAAI2022.
>
> [2] Sign is Not a Remedy: Multiset-to-Multiset Message Passing for Learning on Heterophilic Graphs. ICML2024.
>
> > W2: The key insight behind the proposed method requires further elaboration. While the authors claim that labeling subgraphs better captures local homophily distribution, the mechanism by which this information enhances GNN performance remains unclear. The manuscript should explicitly address several key questions. How does the degree of local homophily influence GNN behavior? What specific advantages does this approach offer for fundamental homophily and heterophily GNNs? A detailed theoretical analysis would strengthen the paper's contribution and provide valuable insights for future research directions.
>
> This question is very insightful, we thank you for asking it. Based on your suggestion, we have proved an **additional theorem** about how the local homophily influences GNN behavior. We have included the theorem in the appendix. Specifically, we show that correct local homophily is necessary for high accuracy. And our method KyN, is designed exactly to reveal the correct homophily. From Figure 2 in the original submission, we observe that the KyN-selected training set most accurately reflects the local homophily.
>
> > W3(2): Validation on large-scale datasets to substantiate efficiency claims and demonstrate practical applicability.
>
> Thanks for the suggestion. We have conducted more experiments on a large-scale dataset, Snap-patents. It has 2,923,922 nodes and 13,975,788 edges. We set the budget to $5C$ and compare KyN with previous GALs.
>
> |             | Accuracy | Time (second) |
> |-------------|----------|------|
> | Random      | 32.9     | 0.06 |
> | Uncertainty | 25.5     | 0.45 |
> | Density     | 25.1     | 752  |
> | AGE         | -        | OOT  |
> | ALG         | -        | OOT  |
> | FeatProp    | -        | OOT  |
> | GraphPart   | -        | OOT  |
> | KyN(Ours)   | 33.7     | 651  |
>
> Due to the limited time during the discussion phase, we record GALs exceeding 30 minutes as OOT (out of time). We observe KyN achieve SOTA performance on this large heterophilic graph, and the runtime is reasonable. This experiment on the large graph shows the efficiency of our KyN.

---

> ### Author Response · Authors · 2024-11-19
> **Response to Reviewer L9AD (Part 2)**
>
> > W3(3) & Q2: Detailed ablation studies to isolate the impact of individual components.
>
> We have conducted more ablation studies based on your suggestion. We use two datasets, Roman-empire and Tolokers, with a budget set to $5C$. We also include the naive random sampling for comparison.
>
> |                              | Roman-empire | Tolokers |
> |------------------------------|--------------|----------|
> | KyN                          | 44.8         | 71.0     |
> | KyN w.o. Importance sampling | 43.7         | 68.5     |
> | KyN w.o. Concatenation       | 44.0         | 70.3     |
> | Random                       | 43.1         | 65.4     |
>
> We observe even without the full importance sampling, our model is still better than the naive random sampling. But the performance degrades without the representative information.
>
> > W4 & Q5 & Q6: The presentation requires significant improvement. The core principle of "know your neighbor" lacks precise definition and theoretical grounding. The relationship between local homophily distribution and active learning effectiveness requires clearer articulation. A substantial revision would enhance readability and impact.
>
> We greatly appreciate your insightful questions regarding the presentation of our paper. Your suggestions have been extremely helpful to us. Based on your feedback, we have made several additions to the paper, including:
>
> 1.We have provided a more detailed explanation of the "know your neighbors" principle and **theoretically** (that is the second theorem we add at the discussion stage) demonstrated why it is meaningful for better estimating local homophily.
>
> 2.We have included pseudocode for our algorithm, which also addresses the corner case you mentioned.
>
> 3.We have elaborated further on the computation and time complexity of the $\ell_1$ Lewis weights.
>
> > Q1: The experimental setup in Table 1 requires clarification regarding the feature combination mechanism in the backbone GNN. How are neighbor-aggregated features combined with target node features? Does the choice of combination method impact performance significantly?
>
> As stated in Section 5.1 of the original submission, we use GraphSAGE as the GNN backbone because it shows great performance on heterophilic graphs[1]. GraphSAGE *concatenates* the embedding of each target node and the average of its neighbors, that alows the so-called "negative aggregation". We understand your concerns about the GNN backbone, since the "ego-neighbor separation" is very crucial to heterophilic graphs[1,2]. That is precisely why we use GraphSAGE (to avoid the impact of inappropriate backbones) instead of GCN or GAT. We believe our rationale aligns with your considerations on this matter.
>
> [1] A critical look at the evaluation of GNNs under heterophily: Are we really making progress? ICLR2023.
>
> [2] Beyond homophily in graph neural networks: Current limitations and effective designs. NeurIPS2020.
>
> > Q3: The experimental analysis in Figure 1 should be expanded, for example, by extending the labeling budget range to start from 2c to align with the standard GAL setting.
>
> Based on your suggestion, we have extended the budget range to start from $2C$.
>
> > Q4: What is the performance comparison with a sample-then--hop approach? The terminology also needs revision (sample as verb vs. k-hop as noun).
>
> We have revised the terminology. As stated in the original submission, we believe that "sample than select $k$-hop" is not a good idea, since it tends to select giant connected component. Based on your suggestions we have implemented a simple "sample than select $k$-hop" method that first randomly select nodes and their one-hop neighbors. The results are as follows, the labeling budget is set to $5C$.
>
> |                            | Roman-empire | Amazon-ratings | Tolokers |
> |----------------------------|--------------|----------------|----------|
> | Sample then select $k$-hop | 43.4         | 30.5           | 66.7     |
> | KyN                        | 44.8         | 31.2           | 71.0     |
>
> We observe that the "sample than select $k$-hop" scheme is indeed suboptimal.
>
> We want to thank you again for your valuable suggestions. We have put a lot of effort to solve the problems you mentioned, and we would be very grateful if you could review the new version we uploaded and give us more feedback.

---

> ### Author Response · Authors · 2024-11-25
> **Looking forward to your reply**
>
> Dear Reviewer L9AD,
>
> As we are days away from the closure of the discussion phase, we eagerly await your feedback on the revisions made. We understand that you may have put in a lot of effort during the discussion phase, and we truly appreciate your hard work. As reviewers for the other papers, we responded to the authors promptly. We would greatly appreciate it if you could also take a look at the changes we made.
>
> Your suggestions have been extremely helpful to us. Based on your feedback, we conducted **five additional experiments**, proved **two theorems**, and made several revisions. We believe that the current version is a significant improvement over the initial submission, and we sincerely hope to earn your approval.

---

> > ### Comment · Reviewer_L9AD · 2024-11-25
> > **Official Comment**
> >
> > Thank you for your comprehensive response. Many of my concerns regarding the experiments have been addressed.
> >
> > However, I still have some concerns, outlined below:
> >
> > - Theorem 1 is too straightforward, might not be appropriate to be termed as "Theorem 1".
> > - The assumption regarding data distribution in Theorem 2 is too strong.
> > - Another major concern arises from the discrepancy between model accuracy and local homophily. Given a budget of labeled nodes, the more labeled neighbors node $i$ has, the more precise our understanding of node $i$'s local homophily becomes. This, in turn, assists the model in generating more accurate embeddings for nodes within node $i$'s neighborhood.
> > However, we encounter a knowledge gap concerning nodes like $j$ that lie outside this neighborhood. How to get prior knowledge about node $j$ remains unclear. Furthermore, can the presence of small values of $|\hat{h}_i-h_i|$ in Eq. 7 contribute to improving the value of $\mathcal{D}(h_t,\hat{h}_t)$ in Eq.6 across all nodes?
> > - Although the authors provide the running time on a larger dataset, they do not present results from other baselines. A more suitable approach would be to omit a method if it cannot complete within 1 or 2 days rather than 30 minutes.
> >
> > I appreciate the authors' efforts to enhance the quality of their manuscript. However, I am inclined to uphold my borderline evaluation and can only raise my rating to 5.

---

> > > ### Author Response · Authors · 2024-11-25
> > > **Thank you**
> > >
> > > Dear Reviewer L9AD,
> > >
> > > Thank you so much for increasing the score! You’ve truly made our day with your support.
> > >
> > > We also want to thank you for recognizing our efforts. Below are our responses to the remaining concerns you raised.
> > >
> > > > Theorem 1 is too straightforward, might not be appropriate to be termed as "Theorem 1".
> > >
> > > We will change Theorem 1 to Proposition 1.
> > >
> > > > The assumption regarding data distribution in Theorem 2 is too strong.
> > >
> > > We understand that it is important to keep all assumptions reasonable. By "the assumption regarding data distribution", do you mean the hypergeometric distribution? Theorem 2 says if we select $n_i$ neighbors of node $i$, the estimation of local homophily will be more accurate with a larger $n_i$. In the proof, we let $X$ be the number of nodes that have the same label as $i$. That is equivalent to drawing balls from the urn. Suppose the urn contains $P$ white and $|N(i)|-P$ black balls, we are actually drawing $n_i$ balls and $X$ follows a hypergeometric distribution.
> > >
> > > It must be the notation that makes this a bit confusing here. We will change $X$ to $K$. Sorry for the inconvenience.
> > >
> > > > Another major concern arises from the discrepancy between model accuracy and local homophily. Given a budget of labeled nodes, the more labeled neighbors... in Eq.6 across all nodes?
> > >
> > > We thank you for this insightful question. We greatly appreciate your careful review of the new content we added. From this question, it’s clear that you have a very accurate and deep understanding of our paper. We agree that covering all the nodes in the graph, rather than just the neighborhood of a single node, is very important. Since, as you have precisely pointed out, we need to get prior knowledge about nodes and make sure most nodes have small $|\hat{h}_i-h_i|$. So instead of sampling a few nodes and acquiring their neighbors, we first partition the graph into small subgraphs and sample the subgraphs. This makes it easier to control the (topological) diversity of the selected nodes since we won't quickly exhaust the budget due to neighbor explosion.
> > >
> > > > Although the authors provide the running time on a larger dataset, they do not present results from other baselines. A more suitable approach would be to omit a method if it cannot complete within 1 or 2 days rather than 30 minutes.
> > >
> > > Thank you for your suggestion. We are rerunning the program immediately and will do our best to update the results before the discussion phase ends.
> > >
> > > We would like to express our sincere gratitude once again. Your feedback means a great deal to us. We also wish you the best of luck with your own submission.

---

> > > > ### Comment · Reviewer_L9AD · 2024-11-25
> > > > **Official Comment**
> > > >
> > > > Thank you for your prompt response.
> > > >
> > > > While the current version of the manuscript may not be accepted for ICLR due to *its heuristic nature lacking sufficient theoretical grounding and the challenging unsolved discrepancy*, I still encourage the authors to incorporate these discussions and make efforts to improve the presentation in their revised paper for future submissions. These discussions, along with additional experiments, have significantly enhanced the quality of this paper. Additionally, the problem you have studied is indeed important.

---

> > > > > ### Author Response · Authors · 2024-11-26
> > > > > **Thank you**
> > > > >
> > > > > Dear Reviewer L9AD,
> > > > >
> > > > > Thank you for recognizing our research problem and we respect your decision. We will continue to improve our paper based on your suggestions, even if we may not get it accepted this time. We will also incorporate these precious discussions in the manuscript, a final version will be uploaded within one or two days and you can take a look at it if you have some spare time.
> > > > >
> > > > > We wish you good luck in the extended discussion period!
> > > > >
> > > > > Best regards,
> > > > >
> > > > > Authors

---

### Official Review · Reviewer_kfNi · 2024-11-03

**Soundness:** 3
**Presentation:** 4
**Contribution:** 3
**Rating:** 6
**Confidence:** 2

**Summary:**

This paper explores the reason for the failure of active learning on heterophilic graphs -- selected sets show homophily. They argue that this is because nodes are queried without knowing neighbors, leading to isolated and homophilic nodes. With this motivation, they propose partitioning nodes into subgraph groups and designing representation using the Jordan Center. Through sampling subgraphs with probabilities proportional to l1 Lewis weights and adding all nodes in subgraphs to the training set, this paper provides guarantees over the gap between the objective function of the training set and the overall graph.

**Strengths:**

- The presentation and storytelling of this paper are smooth and attractive.
- The phenomenon of GAL failing in heterogeneous graphs is first revealed and researched.
- Experiment result improvements are significant and convincing.

**Weaknesses:**

- Please see questions.
- To support your claim, it would be better to provide a case study figure over selected nodes for different methods on a small graph, besides a toy example.

**Questions:**

- In Tab 1. Texas 20C, Uncertainty reaches 94.7 ± 1.4, while KyN is 93.2 ± 1.6. Is it a typo?
- In line 185: "When fed to GNNs, these isolated nodes are viewed as strongly homophilic nodes...". Can you clarify if it is true only for normal homophily-based GNNs, or the same for heterogeneous GNNs?

---

> ### Author Response · Authors · 2024-11-19
> **Response to Reviewer kfNi**
>
> Dear Reviewer kfNi,
>
> We appreciate your insightful review, your positive feedback means a lot to us. Below are our responses.
>
> > Q1: In Tab 1. Texas 20C, Uncertainty reaches 94.7 ± 1.4, while KyN is 93.2 ± 1.6. Is it a typo?
>
> Thank you for your reminder. We have fixed the typo.
>
> > Q2: In line 185: "When fed to GNNs, these isolated nodes are viewed as strongly homophilic nodes...". Can you clarify if it is true only for normal homophily-based GNNs, or the same for heterogeneous GNNs?
>
> We thank you for this insightful question. These isolated nodes are viewed as homophilic nodes by both normal GNNs and heterophilic GNNs. A major difference of normal GNNs and heterophilic GNNs is the *ability* to learn negative aggregation, but if the GAL-selected training set does not reveal heterophily, even heterophilic GNNs cannot learn the negative aggregation. We also conducted more experiments based on other reviewers' suggestions. The following is the result of two heterophilic GNNs, FAGCN [1] and M2M-GNN [2] on Roman-empire with the labeling budget of $20C$.
>
> |             | FAGCN          | M2M-GNN        |
> |-------------|----------------|----------------|
> | Random      | 52.0 $\pm$ 0.5 | 58.3 $\pm$ 1.3 |
> | Uncertainty | 47.7 $\pm$ 1.8 | 54.9 $\pm$ 1.0 |
> | Density     | 45.3 $\pm$ 1.1 | 51.5 $\pm$ 1.2 |
> | AGE         | 50.5 $\pm$ 1.9 | 55.7 $\pm$ 1.4 |
> | ALG         | 51.2 $\pm$ 1.3 | 56.3 $\pm$ 1.2 |
> | FeatProp    | 51.7 $\pm$ 0.9 | 57.0 $\pm$ 0.8 |
> | GraphPart   | 51.6 $\pm$ 1.2 | 56.1 $\pm$ 0.8 |
> | KyN(Ours)   | 53.5 $\pm$ 1.4 | 59.2 $\pm$ 1.0 |
>
> We observe that on these two heterophilic GNNs, our KyN also selects valuable training set, which further supports our argument.
>
> [1] Beyond low-frequency information in graph convolutional networks. AAAI2022.
>
> [2] Sign is Not a Remedy: Multiset-to-Multiset Message Passing for Learning on Heterophilic Graphs. ICML2024.
>
> > W2: To support your claim, it would be better to provide a case study figure over selected nodes for different methods on a small graph, besides a toy example.
>
> Thanks for this interesting suggestion! We have added this case study figure to our appendix.
>
> We want to thank you again for your valuable comments! We have uploaded the revised paper, please feel free to check it and we are looking forward to discussing it further with you.

---

> > ### Comment · Reviewer_kfNi · 2024-11-22
> >
> > Thank you for your responses. I appreciate your effort and I keep my positive score.

---

> > > ### Author Response · Authors · 2024-11-22
> > > **Thank you**
> > >
> > > We sincerely thank you for supporting our work.

---

### Author Response · Authors · 2024-11-22
**Looking forward to your reply**

We thank the reviewers for their insights and constructive suggestions. We have addressed your concerns and revised the paper. The changes made in the manuscript are highlighted in red font. The major additional changes are listed below.

**Two additional theoretical analyses:** Following your suggestions, we have proved two new theorems. The first one says correct local homophily is necessary for high accuracy, which supports our analysis of GAL performance by analyzing local homophily. The second one says our principle of "know your neighbors" helps us better estimate local homophily.

**Comprehensive additional experiments:** Following your suggestions, we have conducted three types of additional experiments, including (1) GAL with different backbones, (2) experiments on a very large graph (with 2M nodes), and (3) more ablation studies. We have invested a significant amount of time and effort into conducting these experiments.

**Improvement in presentation:** Following your suggestions, we have revised our paper to improve readability. In particular, we added a case study figure of GAL-selected training sets, added pseudocode for our algorithm, added the description of computation and the time complexity of the $\ell_1$ Lewis weights, and added a new plot of the node homophily distribution.

We hope these revisions will satisfactorily address the concerns raised by the reviewers and improve the overall quality of our work. We would appreciate it if you could provide more feedback on this revised version. Thanks for your time and effort.

---

### Author Response · Authors · 2024-11-28
**The final version has been uploaded**

Dear reviewers,

We have uploaded the final version of our paper. Compared to the previous version, we made the following changes:

**Modify the notation and add more discussion of theory:** Based on the insightful feedback from Reviewer L9AD, we have modified some notations to better show that there are no strong assumptions or theoretical gaps. Since we are not working on an ideal setup with an infinite labeling budget, it is not possible to make all nodes in a graph "know" their neighbors or get the right label (otherwise we do not need to train a GNN). We believe this situation is natural due to the practical constraints, and it is not a flaw of our method or theory. We thank Reviewer L9AD for these profound questions, and the discussions are incorporated in Appendix A and B.

**Incorporate all experimental results during the rebuttal stage to improve the completeness:** Based on the suggestion of Reviewer LKPL and Reviewer L9AD, we further improve the completeness of our paper by incorporating all experimental results during the rebuttal stage. This includes Table 2 and 3 in the main text, Table 5, 6, and 7 in the Appendix. Based on the suggestion of Reviewer L9AD, we extended the OOT threshold in Table 3 to 24 hours, and we got the results of some baselines.


**Provide the aggregation formula of GraphSAGE:** We added the formula of GraphSAGE in Section 5.1 for readers who are not familiar with this encoder.

Compared to the original submission, we did not change any major conclusions in the paper. We added a lot of content because the comments were very helpful to us. We strictly followed the conference guidelines in revising the paper (e.g., "The revised version shouldn't read like a different paper compared to your original abstract submission").

We want to thank you again for your time and effort in supporting our paper and this conference. We would be very grateful if you could take a look at our final version and tell us what you think. **We also fully understand that you need to prioritize your own submissions at this moment.** We wish all of you good luck at the extended discussion stage.

Best regards,

Authors

---

### Meta-Review · Area_Chair_PPSK · 2024-12-17

**Metareview:**

This paper proposes an approach called KyN (Know Your Neighbors) which aims to improve the state of graph active learning (GAL) on heterophilous graphs.  The authors observe that existing GAL approaches do not generalize well to heterophilous graphs and imply homophilous priors.  The proposal involves selecting nodes based on information about their neighbors through partitioning and an appropriate sampling strategy.

Reviewers were largely borderline on this work, but leaned slightly negatively, hence the reject recommendation.  There were a few weaknesses raised that are worth considering in the revision:

-  Some of theoretical results appear to rely on overly strong assumptions and appear trivial to reviewers (L9AD)

- Reviewers felt the implications for the method and its performance on homophilous graphs were glossed over (kN83, AM9W)

- There are some concerns around the justification of using METIS and the impact of this proposal on the method's effectiveness (kN83, LKPL)

- Some experimental details required to reproduce the empirical results are missing and required some prompting to add inot the paper (LKPL)

The authors made considerable efforts during the rebuttal to address concerns, and strengthened their work considerably.  Still, there is room for improvement and I encourage the use of this feedback for revision.

**Additional Comments On Reviewer Discussion:**

Reviewers raised several concerns, some of which are listed in the above response.  Several points which are not discussed above are around clarifications around benchmark choice (e.g. the use of GraphSAGE), and certain adopted definitions (homophily).

Authors provided several new results including two theorems, as well as additional empirical results on other heterophilic GNN and MLP benchmarks during the rebuttal, as well as an experiment showing that results hold positively with the use of other clustering methods besides METIS. Several clarifications were also made around positioning of theoretical work.  Overall, a few reviewres raised their scores, but maintained assessments around the borderline without strong positive support.

I believe the authors can fold the results into their final submission, as well as incorporate feedback that was unaddressable during the rebuttal to make a more compelling next-submission.

---

### Decision · Program_Chairs · 2025-01-22

Reject